# Adaptive Robust Autonomous Obstacle Traversal Controller for Novel Six-Track Robot

**Rengui Bai** [1,2]**, Runxin Niu** [1]**, Jie Wang** [1,*] **and Zhaosheng Xu** [1]

1   Hefei Institutes of Physical Science, Chinese Academy of Sciences, Hefei 230031, China
2   Science Island Branch, University of Science and Technology of China, Hefei 230026, China
\*   Correspondence: jwang@hfcas.ac.cn; Tel.: +86-152-5511-9227

**Abstract:** The separate control method of flippers and the movement of the mass center makes the active articulated tracked robot unable to realize higher-level motion and difficult to adapt to rough and complex obstacle terrain. In this paper, a new design method of distributed autonomous obstacle traversal controller for the novel six-track robot is proposed. The controller establishes a unified control framework that includes all degrees of freedom of the robot so that the center of mass tracking error and flipper motion tracking error can converge simultaneously to achieve obstacle traversal independently of specific terrain or tasks. First, the forward kinematics model and differential kinematics model of tracked robot are established to generate 3D motion, including flipper angular velocity and body traction velocity. Then, the differential drive robot model is extended into the differential kinematic model to eliminate the slip effect during obstacle traversal. Finally, the feedback control law of the control system and the optimal solution for the singular position of the robot structure are established. In addition, several simulation experiments and physical prototype experiments in different obstacle terrains are executed. In the virtual simulation experiment, the average trajectory error of the flipper is about 0.029 m. In the physical prototype experiment, compared to the manual remote controller and the prior art controller, the average error norm of the center of mass is reduced by 40.7% and 13.5%, respectively; the maximum slip norm is reduced by 34.6% and 19.9%, respectively; and the obstacle crossing time is reduced by 21.3% and 9.3%, respectively, and they validate the accuracy and effectiveness of the designed controller.

**Keywords:** tracked robot; rough terrain; active flippers; differential kinematic model; adaptive robust controller

## 1. Introduction

Currently, tracked robots are increasingly used in firefighting and rescue, post-disaster rescue [1,2], planetary exploration [3–6], and in biochemical contamination scenarios, where the terrain conditions are usually complex and unpredictable. Compared to wheeled and legged robots, tracked robots are more suitable for this type of terrain because the larger contact area of the tracks with the terrain allows for better load and traction capacity. Moreover, this terrain requires robots with a high degree of mobility to overcome obstacles, a capability that depends on the robot's mechanical design. Therefore, researchers have made many improvements to the mechanical structure of robots, typically adding appropriate degrees of freedom at the legs [7], wheels [8], or track joints [9,10]. However, a tracked robot with a high degree of freedom presents higher operation requirements for the remote operator, while the increase in cognitive load inhibits the operator from performing higher-level tasks. In particular, manual operation becomes extremely difficult when the teleoperator is traversing rough terrain with limited visual feedback that blurs and flickers.

In order to improve the autonomous ability and terrain adaptability of tracked robots, researchers have extensively studied the issues of stability [11], self-reconfiguration [12–14], track-terrain interaction [15,16], and control [17–19].

Nagatani, Okada, et al. [20] focused on active flipper control of the tracked robot Kenaf and its second-generation Quince, leading to the development of a shared autonomy system, which consists of a remotely controllable main track controller and a fully autonomous sub-track controller. The autonomous system allows unskilled operators to traverse rough terrain by remotely controlling the main track. However, the developed shared autonomous system was not based on the forward and differential kinematic models, whereas the separate control of the main track and the sub-tracks enhances the influence of the main track speed on the obstacle traversal by the sub-tracks. Chen et al. [21,22] proposed a composite motion mode for a tracked robot octopus with four arms and four flippers. The presented control system is able to accurately estimate terrain information based on joint angle and torque data, which are used to implement control of flippers, tracks, and arms. The combined crawling and walking motion mode provides higher terrain adaptability in complex unstructured terrain scenarios. Suzuki et al. [23,24] developed a robotic system that can autonomously climb and descend stairs. The three-dimensional (3D) scanning device installed on the Meisei Rescue Mk-4 robot is used for obstacle detection and flipper configuration based on the type of the detected obstacle, while the IMU is used for trajectory tracking control. However, the terrain environment in rescue scenarios is often unpredictable and complex, while the simple flipper configuration cannot meet the requirements of real rescue tasks. Colas et al. [25] proposed an autonomous stair climbing method, based on path planning, in which the body and flipper paths are planned separately, making the control of the flipper independent of the center of mass controller. In addition, the flipper configuration does not consider the influence of the terrain on the slip, which introduces posture estimation inaccuracy, causing the flipper to execute an incorrect adjustment. Kojima et al. [26] proposed an autonomous spiral stair-climbing method, using the reaction force from side walls by installing passive wheels at collision points and automatically controlling the main track and sub-tracks. Geometric models are used to confirm the trajectory convergence of the tracked robot during stair climbing; however, the scheme is performed based on a known terrain environment in which collision forces and slips may have an impact on the flipper configuration. Zhang et al. [27] proposed an online method for adjusting the center of mass position of a tracked robot to enhance rollover stability by changing the configuration of the robot arm. The experiment proves that this method increases the stability of the tracked robot when traversing uneven terrain, but it does not consider the effects of collision and slipping between the flipper and the terrain. In fact, the wrong flipper configuration will lead to the inability of the tracked robot to traverse complex terrain as well as affect its stability.

However, most tracked robots with active flippers are usually used in low-speed, small-load rescue scenarios without considering the multiple terrains requirement of adaptability. The research work involves specific tasks or specific terrain, such as stairclimbing in urban ruin scenarios or typical obstacle traversal in wilderness rescue missions [28–30]. In addition, the flipper control scheme of tracked robots is usually configured according to the terrain and is independent of the center of mass controller. The lack of a unified control framework that can include all the degrees of freedom of the tracked robot while considering crawler–terrain interaction and the adaptability of the flipper makes the tracking errors of the centroid controller and the flipper controller fail to converge synchronously, whereas the flipper cannot generate higher-level movements.

Therefore, the main contributions of the paper include:

(1)   A new six-track robot, with wheeled, legged, and tracked advantages, is developed. It is capable of high-speed driving; it has large load and high mobility; and it can adapt to flat, soft, and obstacle-ridden terrain.

(2)   A new distributed autonomous obstacle traversal controller for the six-track robot is designed. The concepts of robotic forward and differential kinematics are applied to the six-track robot and a unified control framework, including all joint degrees of freedom, is established. The controller also integrates a slip-steering model, thus taking into account both the slip effects between the track and the terrain and the adaptation of the flippers.

(3)     The feedback control law of the control system is derived based on the differential kinematic model, which enables the controller to control each flipper individually, achieving obstacle traversal independently of specific terrain and tasks. The optimal solution for the singular position of the six-track robot structure is also derived. Finally, the robustness and effectiveness of the controller are verified via simulation and experiment.

The rest of this paper is organized as follows: Section 2 establishes the forward kinematic model of the six-track robot. The differential kinematic models for the single leg and system of the six-track robot are given in Sections 3 and 4, respectively. Section 5 introduces a unified adaptive controller, while a virtual scenario experiment and a physical prototype experiment are designed in Section 6 to verify the performance of the proposed controller. Finally, the main conclusions are summarized in Section 7.

## 2. Forward Kinematic Model of the Six-Track Robot

In this section, a forward kinematic model of the six-track robot, describing the mapping of the position between the end points of the tracks and the robot body, is developed. The six-track robot can be modelled in a simplified approach as a floating base and six three-degree-of-freedom series kinematic chains, namely a floating base (0), left front (1), right front (2), left center (3), right center (4), left rear (5), and right rear (6). Each kinematic chain contains one passive joint and two active joints. The left kinematic chain is formed by connecting the left track ($link(i, 2), i \in (1, 3, 5)$) to the left cantilever ($link(i, 1), i \in (1, 3, 5)$) through an active rotating joint ($\theta_{i,2}, i \in (1, 3, 5)$). Similarly, the right kinematic chain is formed by connecting the right track ($link(i, 2), i \in (2, 4, 6)$) to the right cantilever ($link(i, 1), i \in (2, 4, 6)$) through an active rotating joint ($\theta_{i,2}, i \in (2, 4, 6)$). It should be noted that the left and right cantilevers are connected to the robot body ($link\ 0$) through the passive rotating joints ($\theta_{i,1}, i \in (1, 2, 3, 4, 5, 6)$). The mechanical structure and the kinematic chain numbering of the six-track robot are shown in Figure 1.

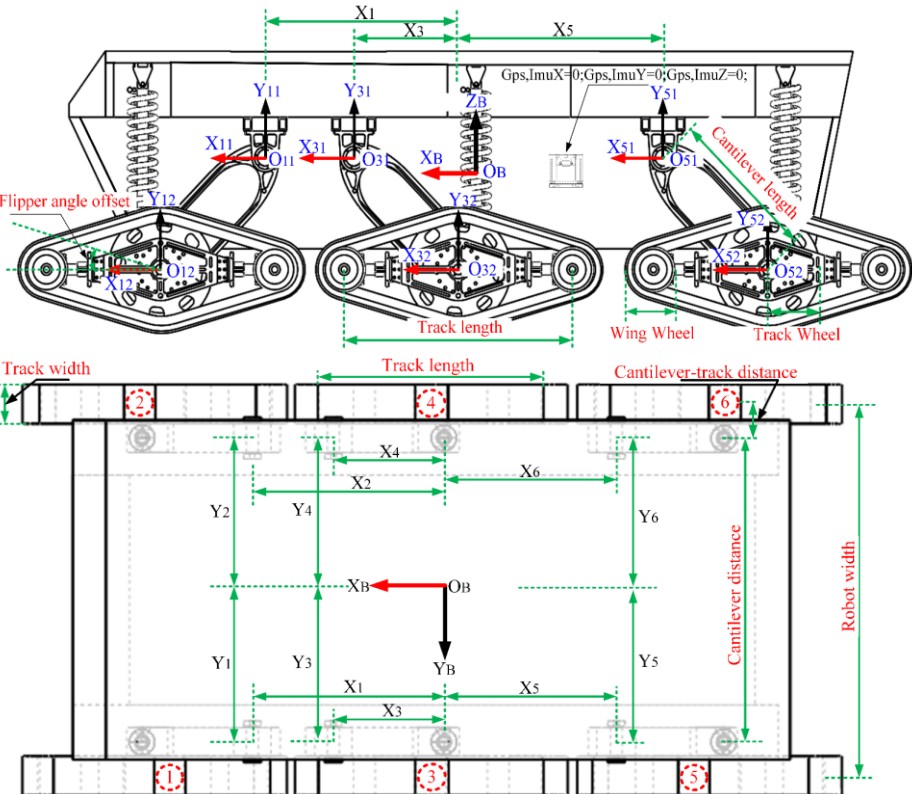

**Figure 1.** Mechanical structure of the six-track robot.

The world coordinates system $\{C_G\}$ is fixed on the ground, and the body coordinates system $\{C_B\}$ and base link coordinates system $\{C_0\}$ are fixed on the center-of-mass of the robot body. The cantilever coordinates system $\{{}^iC_1\}$, the track coordinates system $\{{}^iC_2\}$, and the wing–wheel coordinates system $\{{}^iC_3\}$, which are fixed on the rotating joints, are also established. The $Z_{i,j}$ axis is set as the rotation axis of the joint $(i, j)$, while the $X_{i,j}$ axis is perpendicular to the $Z_{i,j-1}$ and $Z_{i,j}$ axes, with a direction from the $Z_{i,j-1}$ axis to the $Z_{i,j}$ axis, where $i \in (1, 2, 3, 4, 5, 6)$, $j \in (1, 2)$. The $j$th rotation joint of the $i$th leg kinematic chain are denoted as shown in Figure 2. The nominal parameters of the six-track robot are given in Table 1. The Denavit–Hartenberg (D-H) parameters of the kinematic chain are given in Table 2.

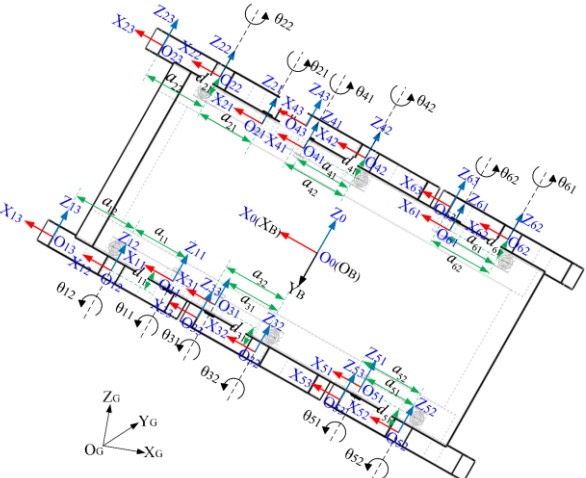

**Figure 2.** Forward kinematic model of the six-track robot.

**Table 1.** Nominal parameters of six-track robot.

| Parameter | Description |
|:---:|:---:|
| $\delta$ | Flipper angle offset |
| $T$ | Track length |
| $W$ | Cantilever distance |
| $D$ | Cantilever–track distance |
| $R$ | Track wheel radius |
| $r$ | Wing wheel radius |
| $L$ | Cantilever length |

**Table 2.** D-H parameters of the six-track robot, where $i \in (1, 2, 3, 4, 5, 6), j \in (1, 2)$.

| Link | $\theta_{i,j}$ | $d_{i,j}$ | $a_{i,j}$ | $\alpha_{i,j}$ |
|:---:|:---:|:---:|:---:|:---:|
| $i, 1$ | $\theta_{i,1}$ | $\left(-1^{i+1}\right)D$ | $L$ | $0$ |
| $i, 2$ | $\theta_{i,2} + \delta$ | $0$ | $T/2$ | $0$ |

${}^{i,1}\mathbf{T}_2$ and ${}^{i,1}\mathbf{T}_3$ denote the transformation matrices of the track coordinates system $\{{}^iC_2\}$ and the wing–wheel coordinates system $\{{}^iC_3\}$, with respect to the cantilever coordinates system $\{{}^iC_1\}$, respectively. The coordinates transformation between the kinematic chains can be calculated using the D-H parameter. The homogeneous transformation matrix of the track coordinates system $\{{}^iC_2\}$, with respect to the cantilever coordinates system $\{{}^iC_1\}$, in the $i$th leg kinematic chain can be expressed as:

$$
{}^{i,1}\mathbf{T}_2(\theta_{i,1}) = \begin{bmatrix} {}^{i,1}\mathbf{R}_2(\theta_{i,1}) & {}^{i,1}\mathbf{P}_2(\theta_{i,1}) \\ \mathbf{0}^T & 1 \end{bmatrix} = \begin{bmatrix} c_1 & -s_1 & 0 & Lc_1 \\ s_1 & c_1 & 0 & Ls_1 \\ 0 & 0 & 1 & D \\ 0 & 0 & 0 & 1 \end{bmatrix} \tag{1}
$$

$L$ is the vertical distance between the cantilever coordinates system and the track coordinates system about the $Z$ axis that is the length of the cantilever. $D$ is the offset distance between the cantilever coordinates system and the track coordinates system. The homogeneous transformation matrix of the wing–wheel coordinates system $\{^{i}C_3\}$ with respect to the track coordinates system $\{^{i}C_2\}$ can be expressed as:

$$
^{i,2}\mathbf{T}_3(\theta_{i,2}+\delta) = \begin{bmatrix} ^{i,2}\mathbf{R}_3(\theta_{i,2}+\delta) & ^{i,2}\mathbf{P}_3(\theta_{i,2}+\delta) \\ \mathbf{0}^T & 1 \end{bmatrix} = \begin{bmatrix} c_2 & -s_2 & 0 & \frac{T}{2}c_2 \\ s_2 & c_2 & 0 & \frac{T}{2}s_2 \\ 0 & 0 & 1 & 0 \\ 0 & 0 & 0 & 1 \end{bmatrix} \tag{2}
$$

where $T$ is the distance between the centers of the two wing wheels of the track and $\delta$ is the flipper angle offset. Let $\bar{\bar{\boldsymbol{\theta}}}_i = (\theta_{i,1}, \theta_{i,2}+\delta)^T$ be the joint configuration vector of the $i$th leg kinematic chain, while the homogeneous transformation matrix of the wing–wheel coordinates system with respect to the base link coordinates system $\{C_0\}$ can be expressed by Equations (1) and (2).

$$
^{i,0}\mathbf{T}_3\left(\bar{\bar{\boldsymbol{\theta}}}_i\right) = {^{i,0}\mathbf{P}_1} + {^{i,1}\mathbf{T}_2(\theta_{i,1})^{i,2}\mathbf{T}_3(\theta_{i,2}+\delta)} = \begin{bmatrix} c_1c_2 - s_1s_2 & -c_1s_2 - c_2s_1 & 0 & \frac{T}{2}c_1c_2 - \frac{T}{2}s_1s_2 + Lc_1 + x_i \\ c_1s_2 + c_2s_1 & c_1c_2 - s_1s_2 & 0 & \frac{T}{2}s_1c_2 + \frac{T}{2}c_1s_2 + Ls_1 + y_i \\ 0 & 0 & 1 & D + z_i \\ 0 & 0 & 0 & 1 \end{bmatrix} \tag{3}
$$

where $^{i,0}\mathbf{P}_1 = (x_i, y_i, z_i)$ is the position of the cantilever coordinates system with respect to the body coordinates system of the $i$th leg kinematic chain. The symbols $s_1$, $c_1$, $s_2$, and $c_2$ denote $sin\theta_{i,1}$, $cos\theta_{i,1}$, $sin(\theta_{i,2}+\delta)$, and $cos(\theta_{i,2}+\delta)$, respectively. All joint angles are considered positive in the counterclockwise direction. The choice of the base link coordinates system forces the coordinates system $\{C_0\}$ to be rotated clockwise by $\pi/2$ with respect to the body coordinates system $\{C_B\}$, as shown in Figure 2. The constant rotation matrix $^{B}\mathbf{R}_0$ is used to transform the base link coordinates system into the body coordinates system. Then, the homogeneous transformation matrix of the wing–wheel coordinates system under the body coordinates system $\{C_B\}$ can be expressed by:

$$
^{i,B}\mathbf{T}_3\left(\bar{\bar{\boldsymbol{\theta}}}_i\right) = \begin{bmatrix} ^{B}\mathbf{R}_0 & ^{B}\mathbf{P}_0 \\ \mathbf{0}^T & 1 \end{bmatrix} {^{i,0}\mathbf{T}_3\left(\bar{\bar{\boldsymbol{\theta}}}_i\right)}
$$

$$
^{B}\mathbf{R}_0 = \begin{bmatrix} 1 & 0 & 0 \\ 0 & 0 & -1 \\ 0 & 1 & 0 \end{bmatrix} \quad {^{B}\mathbf{P}_0} = \begin{bmatrix} 0 \\ 0 \\ 0 \end{bmatrix} \tag{4}
$$

Let $^{i,3}\mathbf{P}_e$ be the position vector of the track end point with respect to the wing–wheel coordinates system $\{^{i}C_3\}$, whereas the posture relationship of the track end point with respect to the body coordinates system $\{C_B\}$ can be expressed by a function containing the joint angle variables $\theta_{i,1}$ and $\theta_{i,2}$, whose forward kinematic equations are given by the following homogeneous matrix:

$$
\begin{bmatrix} ^{i,B}\mathbf{P}_e \\ 1 \end{bmatrix} = {^{i,B}\mathbf{T}_3\left(\bar{\bar{\boldsymbol{\theta}}}_i\right)} \begin{bmatrix} ^{i,3}\mathbf{P}_e \\ 1 \end{bmatrix} \tag{5}
$$

The mapping between the vector $^{i,3}\mathbf{P}_e$ and the vector $^{i,B}\mathbf{P}_e$ is defined in Equation (5), allowing us to build the relationship between the linear velocity of the track end point and both the angular velocity of the active joint $\dot{\theta}_{i,2}$ and the passive joint $\dot{\theta}_{i,1}$.

## 3. Differential Kinematic Model for the Single Leg of the Six-Track Robot

In this section, the Jacobi matrix for the kinematic chain of Leg 1 (i.e., the left front leg) of the six-track robot will be calculated in detail, while the kinematic chains of the remaining legs are calculated in the same way as Leg 1. The vectors among the left front legs of the tracked robot are illustrated in Figure 3. $^{1,0}\mathbf{P}_1$ and $^{1,0}\mathbf{P}_3$ denote the position vectors of the coordinates system $\{^{1}C_1\}$ and coordinates system $\{^{1}C_3\}$ origin, relative to the coordinates system $\{C_0\}$, whereas the left superscript indicates the kinetic chain of Leg 1. $^{1,1}\mathbf{P}_2$ denotes the position vector of the origin of the coordinates system $\{^{1}C_2\}$ relative to the coordinates system $\{^{1}C_1\}$, while $^{1,2}\mathbf{P}_3$ denotes the position vector of the origin of the coordinates system $\{^{1}C_3\}$ relative to the coordinates system $\{^{1}C_2\}$. The geometric relationship can be obtained as:

$$^{1,0}\mathbf{P}_3 = {}^{1,0}\mathbf{P}_1 + {}^{1,0}\mathbf{R}_1(0)^{1,1}\mathbf{P}_2 + {}^{1,0}\mathbf{R}_1(0)^{1,1}\mathbf{R}_2(\theta_{1,1})^{1,2}\mathbf{P}_3 \tag{6}$$

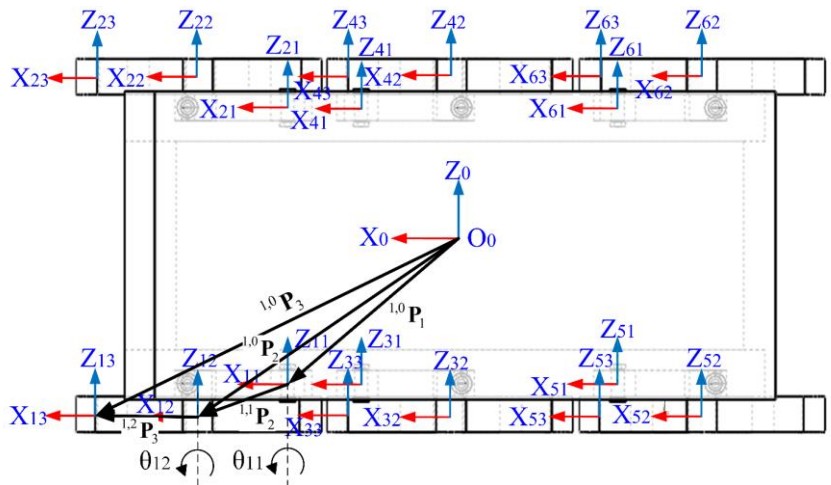

**Figure 3.** Single-leg differential kinematics of six-track robot.

The position between the coordinates system $\{^{1}C_1\}$ and the coordinates system $\{C_0\}$ is fixed without rotational transformation. $^{1,0}\mathbf{P}_1$ and $^{1,0}\mathbf{R}_1(0)$ are constant matrices. Thus, $^{1,0}\mathbf{R}_1(0) \in SO(3)$ is expressed as:

$$^{1,0}\mathbf{R}_1(0) = \begin{bmatrix} 1 & 0 & 0 \\ 0 & 1 & 0 \\ 0 & 0 & 1 \end{bmatrix} \tag{7}$$

Equation (6) is differentiated with respect to time and the constant matrix term is zero. Next, the expression of the linear velocity of Linkage 2 is established by:

$$^{1,0}\dot{\mathbf{P}}_3 = {}^{1,0}\mathbf{R}_1(0)^{1,1}\dot{\mathbf{P}}_2 + {}^{1,0}\mathbf{R}_1(0)^{1,1}\dot{\mathbf{R}}_2(\theta_{1,1})^{1,2}\mathbf{P}_3 + {}^{1,0}\mathbf{R}_1(0)^{1,1}\mathbf{R}_2(\theta_{1,1})^{1,2}\dot{\mathbf{P}}_3 \tag{8}$$

where $^{1,1}\dot{\mathbf{P}}_2$ is the linear velocity of the origin of the coordinates system $\{^{1}C_2\}$, with respect to the coordinates system $\{^{1}C_1\}$, expressed in the coordinates system $\{^{1}C_1\}$. Since Joint 1 is a rotational joint, $^{1,1}\dot{\mathbf{P}}_2$ can be calculated by:

$$^{1,1}\dot{\mathbf{P}}_2 = {}^{1,1}\boldsymbol{\omega}_2 \times {}^{1,1}\mathbf{P}_2 \tag{9}$$

where $^{1,1}\boldsymbol{\omega}_2$ is the angular velocity of the origin of the coordinates system $\{^{1}C_2\}$, relative to the coordinates system $\{^{1}C_1\}$, expressed in the coordinates system $\{^{1}C_1\}$. Since $^{1,1}\boldsymbol{\omega}_2$

and $^{1,1}\mathbf{P}_2$ are not easily observable in the coordinates system $\{^1C_1\}$, Equation (9) is defined in the coordinates system $\{C_0\}$ as follows:

$$^{1,1}\dot{\mathbf{P}}_2 = {^{1,1}}\mathbf{R}_0 \left( {^1}\boldsymbol{\omega}_{1,2}^0 \times {^1}\mathbf{P}_{1,2}^0 \right) \tag{10}$$

where $^1\boldsymbol{\omega}_{1,2}^0$ is the angular velocity of the origin of the coordinates system $\{^1C_2\}$, with respect to the coordinates system $\{^1C_1\}$, expressed in the coordinates system $\{C_0\}$. Similarly, the second term on the right-hand side of Equation (8) is described as:

$$^{1,0}\mathbf{R}_1(0)^{1,1}\dot{\mathbf{R}}_2(\theta_{1,1})^{1,2}\mathbf{P}_3 = {^1}\boldsymbol{\omega}_{1,2}^0 \times \left( {^{1,0}}\mathbf{P}_3 - {^{1,0}}\mathbf{P}_2 \right) \tag{11}$$

Since Joint 2 is a rotational joint, the third term on the right-hand side of Equation (8) can be expressed as:

$$^{1,0}\mathbf{R}_1(0)^{1,1}\mathbf{R}_2(\theta_{1,1})^{1,2}\dot{\mathbf{P}}_3 = {^1}\boldsymbol{\omega}_{2,3}^0 \times \left( {^{1,0}}\mathbf{P}_3 - {^{1,0}}\mathbf{P}_2 \right) \tag{12}$$

Using Equations (9)–(12) in Equation (8), the following relation is derived:

$$^{1,0}\dot{\mathbf{P}}_3 = {^1}\boldsymbol{\omega}_{1,2}^0 \times \left( {^{1,0}}\mathbf{P}_3 - {^{1,0}}\mathbf{P}_1 \right) + {^1}\boldsymbol{\omega}_{2,3}^0 \times \left( {^{1,0}}\mathbf{P}_3 - {^{1,0}}\mathbf{P}_2 \right) \tag{13}$$

where

$$^1\boldsymbol{\omega}_{1,2}^0 = \dot{\theta}_{1,1}{^1}\hat{\mathbf{Z}}_1$$

$$^1\boldsymbol{\omega}_{2,3}^0 = \dot{\theta}_{1,2}{^1}\hat{\mathbf{Z}}_2 \tag{14}$$

where $^1\hat{\mathbf{Z}}_1$ and $^1\hat{\mathbf{Z}}_2$ are the rotation axes of the coordinates system $\{^1C_1\}$ and the coordinates system $\{^1C_2\}$ in the kinematic chain of Leg 1, respectively. Linkage 1 and Linkage 2 rotate at angular velocities of $\dot{\theta}_{1,1}$ and $\dot{\theta}_{1,2}$, respectively. Eventually, Equation (8) can be expressed as:

$$^{1,0}\dot{\mathbf{P}}_3 = \left[ {^1}\hat{\mathbf{Z}}_1 \times \left( {^{1,0}}\mathbf{P}_3 - {^{1,0}}\mathbf{P}_1 \right) \quad {^1}\hat{\mathbf{Z}}_2 \times \left( {^{1,0}}\mathbf{P}_3 - {^{1,0}}\mathbf{P}_2 \right) \right] \begin{bmatrix} \dot{\theta}_{1,1} \\ \dot{\theta}_{1,2} \end{bmatrix} \tag{15}$$

Since the position vector of the track end point with respect to the coordinates system $\{^1C_3\}$ is constant, the Jacobi matrix $\mathbf{J}_1 \left( \bar{\bar{\boldsymbol{\theta}}}_1 \right)$ of the linear velocity of the track end point in the body coordinates system $\{C_B\}$ is computed as follows:

$$^{1,B}\dot{\mathbf{P}}_e = {^B}\mathbf{R}_0{^{1,0}}\dot{\mathbf{P}}_3 = {^B}\mathbf{R}_0 \left[ {^1}\hat{\mathbf{Z}}_1 \times \left( {^{1,0}}\mathbf{P}_3 - {^{1,0}}\mathbf{P}_1 \right) \quad {^1}\hat{\mathbf{Z}}_2 \times \left( {^{1,0}}\mathbf{P}_3 - {^{1,0}}\mathbf{P}_2 \right) \right] \begin{bmatrix} \dot{\theta}_{1,1} \\ \dot{\theta}_{1,2} \end{bmatrix} \tag{16}$$

Considering the forward kinematic model as developed in Section 2, the terms of the Jacobi matrix in Equation (16) can be expressed as:

$$^1\hat{\mathbf{Z}}_1 = {^{1,0}}\mathbf{R}_1(0) \begin{bmatrix} 0 \\ 0 \\ 1 \end{bmatrix}, \quad {^1}\hat{\mathbf{Z}}_2 = {^{1,0}}\mathbf{R}_1(0)^{1,1}\mathbf{R}_2(\theta_{1,1}) \begin{bmatrix} 0 \\ 0 \\ 1 \end{bmatrix}$$

$$\begin{bmatrix} {^{1,0}}\mathbf{P}_1 \\ 1 \end{bmatrix} = \begin{bmatrix} x_1 \\ y_1 \\ z_1 \\ 1 \end{bmatrix}, \quad \begin{bmatrix} {^{1,0}}\mathbf{P}_2 \\ 1 \end{bmatrix} = {^{1,0}}\mathbf{T}_2(\theta_{1,1}) \begin{bmatrix} 0 \\ 0 \\ 0 \\ 1 \end{bmatrix}, \quad \begin{bmatrix} {^{1,0}}\mathbf{P}_3 \\ 1 \end{bmatrix} = {^{1,0}}\mathbf{T}_3 \left( \bar{\bar{\boldsymbol{\theta}}}_1 \right) \begin{bmatrix} 0 \\ 0 \\ 0 \\ 1 \end{bmatrix} \tag{17}$$

Ultimately, the track end point linear velocity in the Leg 1 kinematic chain of the six-track robot can be expressed in the body coordinates system as:

$$^{1,B}\dot{\mathbf{P}}_e = \mathbf{J}_1\left(\bar{\bar{\theta}}_1\right)\dot{\bar{\bar{\theta}}}_1 \tag{18}$$

where $\mathbf{J}_1\left(\bar{\bar{\theta}}_1\right)$ is the Jacobi matrix of the kinetic chain of Leg 1, while the terms on the right-hand side of the equation are as follows:

$$\mathbf{J}_1\left(\bar{\bar{\theta}}_1\right) = \begin{bmatrix} -\frac{T}{2}s_1c_2 - \frac{T}{2}c_1s_2 - Ls_1 & -\frac{T}{2}s_1c_2 - \frac{T}{2}c_1s_2 \\ 0 & 0 \\ \frac{T}{2}c_1c_2 - \frac{T}{2}s_1s_2 + Lc_1 & \frac{T}{2}c_1c_2 - \frac{T}{2}s_1s_2 \end{bmatrix}$$

$$\bar{\bar{\theta}}_1 = \begin{bmatrix} \theta_{1,1} \\ \theta_{1,2} + \delta \end{bmatrix}, \quad \dot{\bar{\bar{\theta}}}_1 = \begin{bmatrix} \dot{\theta}_{1,1} \\ \dot{\theta}_{1,2} \end{bmatrix} \tag{19}$$

The calculation process of the remaining kinematic chains is similar to Leg 1. The linear velocity of each robot track end point $^{i,B}\dot{\mathbf{P}}_e$ can be expressed as:

$$^{i,B}\dot{\mathbf{P}}_e = \mathbf{J}_i\left(\bar{\bar{\theta}}_i\right)\dot{\bar{\bar{\theta}}}_i$$

$$\bar{\bar{\theta}}_i = \begin{bmatrix} \theta_{i,1} \\ \theta_{i,2} + \delta \end{bmatrix}, \quad \dot{\bar{\bar{\theta}}}_i = \begin{bmatrix} \dot{\theta}_{i,1} \\ \dot{\theta}_{i,2} \end{bmatrix} \tag{20}$$

where $\mathbf{J}_i\left(\bar{\bar{\theta}}_i\right)$ is the ($3 \times 2$) geometric Jacobi matrix of the $i$th leg kinematic chain, which establishes the mapping relationship between the linear velocity of the track end point $^{i,B}\dot{\mathbf{P}}_e$ and the joint angular velocities $\dot{\theta}_{i,1}$ and $\dot{\theta}_{i,2}$.

## 4. Differential Kinematic Model for the Six-Track Robot System

In this section, the differential kinematic model of the six-track robot is introduced. Since the tracked robot has six identical serial kinematic chains, only the detailed geometric Jacobi matrix calculation process for the Leg 1 kinematic chain (left front leg kinematic chain) is provided.

Let $^{1,B}\mathbf{P}_e$ be the position vector of the track end point of the Leg 1 kinematic chain in the body coordinates system $\{C_B\}$, while the position vector $^{1,G}\mathbf{P}_e$ of the track end point in the world coordinates system $\{C_G\}$ can be obtained by the following coordinates transformation (as shown in Figure 4):

$$^{1,G}\mathbf{P}_e = {}^{G}\mathbf{P}_B + {}^{G}\mathbf{R}_B \, {}^{1,B}\mathbf{P}_e \tag{21}$$

where $^{G}\mathbf{R}_B$ denotes the rotation matrix of the body coordinates system $\{C_B\}$ with respect to the world coordinates system $\{C_G\}$. $^{G}\mathbf{P}_B$ denotes the position vector of the body coordinates system origin $\{C_B\}$ in the world coordinates system $\{C_G\}$, while differentiating Equation (21) with respect to time yields:

$$^{1,G}\dot{\mathbf{P}}_e = {}^{G}\dot{\mathbf{P}}_B + {}^{G}\dot{\mathbf{R}}_B \, {}^{1,B}\mathbf{P}_e + {}^{G}\mathbf{R}_B \, {}^{1,B}\dot{\mathbf{P}}_e \tag{22}$$

where, $^{G}\dot{\mathbf{P}}_B$ represents the linear velocity vector of the body coordinates system $\{C_B\}$, with respect to the world coordinates system $\{C_G\}$, whereas $^{1,B}\dot{\mathbf{P}}_e$ represents the linear velocity vector of the track end point of leg 1 kinematic chain, with respect to the body coordinates system $\{C_B\}$, which is a function of joint angular velocities $\dot{\theta}_{1,1}$ and $\dot{\theta}_{1,2}$ (see Equation (18)).

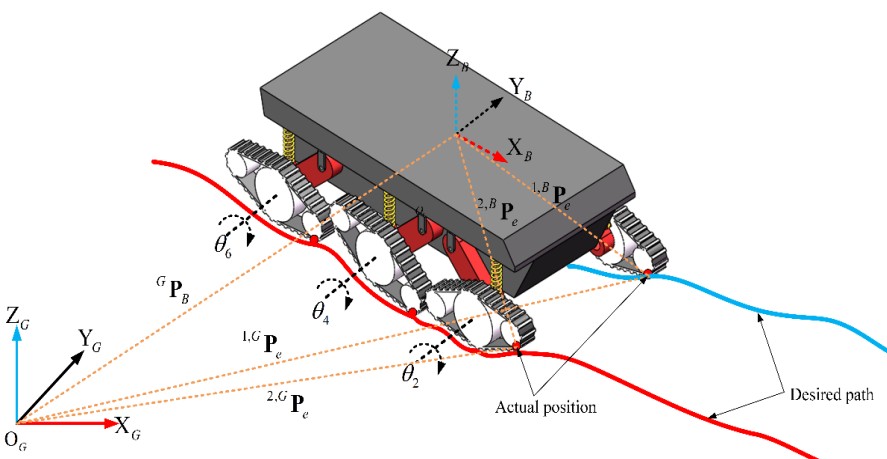

**Figure 4.** Differential kinematics of a six-track robot.

Due to the fact that $^G\mathbf{\Omega}_B$ is the angular velocity vector of the coordinates system $\{C_B\}$ with respect to the coordinates system $\{C_G\}$, the second term on the right-hand side of Equation (22) can be expressed as:

$$^G\dot{\mathbf{R}}_B{}^{1,B}\mathbf{P}_e = [^G\mathbf{\Omega}_B]_\times {}^G\mathbf{R}_B{}^{1,B}\mathbf{P}_e \tag{23}$$

where $[^G\mathbf{\Omega}_B]_\times$ is the antisymmetric matrix about the angular velocity vector $^G\mathbf{\Omega}_B$. According to the formula of antisymmetric matrix and vector product, Equation (23) can be expressed as:

$$[^G\mathbf{\Omega}_B]_\times {}^1\mathbf{P}_{B,e}^G = -\left[^{1,G}\mathbf{P}_e - {}^G\mathbf{P}_B\right] \times {}^G\mathbf{\Omega}_B \tag{24}$$

where $^1\mathbf{P}_{B,e}^G$ represents the position vector of the track end point in the Leg 1 kinematic chain, with respect to the body coordinates system $\{C_B\}$, expressed in the world coordinates system $\{C_G\}$ so that Equation (22) can be expressed as:

$$^{1,G}\dot{\mathbf{P}}_e = {}^G\dot{\mathbf{P}}_B - \left[^{1,G}\mathbf{P}_e - {}^G\mathbf{P}_B\right]_\times {}^G\mathbf{\Omega}_B + {}^G\mathbf{R}_B{}^{1,B}\dot{\mathbf{P}}_e \tag{25}$$

$^G\mathbf{\Omega}_B$ and $[^G\mathbf{\Omega}_B]_\times$ are given as:

$$^G\mathbf{\Omega}_B = \begin{bmatrix} \Omega_x & \Omega_y & \Omega_z \end{bmatrix}^T$$

$$[^G\mathbf{\Omega}_B]_\times = \begin{bmatrix} 0 & -\Omega_z & \Omega_y \\ \Omega_z & 0 & -\Omega_x \\ -\Omega_y & \Omega_x & 0 \end{bmatrix} \tag{26}$$

Equation (25) constructs the mapping relationship between the track end point line velocity $^{1,G}\dot{\mathbf{P}}_e$ expressed in the world coordinates system and the body line velocity $^G\dot{\mathbf{P}}_B$, the body angular velocity $^G\mathbf{\Omega}_B$, and the track end point line velocity $^{1,B}\dot{\mathbf{P}}_e$ expressed in the body coordinates system. $^{1,B}\dot{\mathbf{P}}_e$ is calculated in detail in Section 3 as a function of the angles $\theta_{i,1}$ and $\theta_{i,2}$ and the angular velocities $\dot{\theta}_{i,1}$ and $\dot{\theta}_{i,1}$, given by:

$$^{1,B}\dot{\mathbf{P}}_e = \mathbf{J}_1\left(\bar{\bar{\mathbf{\theta}}}_1\right)\dot{\bar{\bar{\mathbf{\theta}}}}_1 \tag{27}$$

Based on the above derivation, the complete system of differential kinematic equations of the six-track robot can be expressed as:

$$^G\dot{\mathbf{P}}_e = \check{\mathbf{J}}\left(\bar{\bar{\mathbf{\theta}}}_s\right)\dot{\bar{\bar{\mathbf{\theta}}}}_s \tag{28}$$

Among them,

$$^G\dot{\mathbf{P}}_e = \begin{bmatrix} ^{1,G}\dot{\mathbf{P}}_e & ^{2,G}\dot{\mathbf{P}}_e & ^{3,G}\dot{\mathbf{P}}_e & ^{4,G}\dot{\mathbf{P}}_e & ^{5,G}\dot{\mathbf{P}}_e & ^{6,G}\dot{\mathbf{P}}_e \end{bmatrix}^T$$

$$\check{\mathbf{J}}\left(\bar{\bar{\mathbf{\theta}}}_s\right) = \begin{bmatrix} \mathbf{I}_{3\times3} & -\left[^G\mathbf{R}_B, ^{1,B}\mathbf{P}_e\right]_\times & ^G\mathbf{R}_B\mathbf{J}_1\left(\bar{\bar{\mathbf{\theta}}}_1\right) & \mathbf{0}_{3\times2} & \mathbf{0}_{3\times2} & \mathbf{0}_{3\times2} & \mathbf{0}_{3\times2} & \mathbf{0}_{3\times2} \\ \mathbf{I}_{3\times3} & -\left[^G\mathbf{R}_B, ^{2,B}\mathbf{P}_e\right]_\times & \mathbf{0}_{3\times2} & ^G\mathbf{R}_B\mathbf{J}_2\left(\bar{\bar{\mathbf{\theta}}}_2\right) & \mathbf{0}_{3\times2} & \mathbf{0}_{3\times2} & \mathbf{0}_{3\times2} & \mathbf{0}_{3\times2} \\ \mathbf{I}_{3\times3} & -\left[^G\mathbf{R}_B, ^{3,B}\mathbf{P}_e\right]_\times & \mathbf{0}_{3\times2} & \mathbf{0}_{3\times2} & ^G\mathbf{R}_B\mathbf{J}_3\left(\bar{\bar{\mathbf{\theta}}}_3\right) & \mathbf{0}_{3\times2} & \mathbf{0}_{3\times2} & \mathbf{0}_{3\times2} \\ \mathbf{I}_{3\times3} & -\left[^G\mathbf{R}_B, ^{4,B}\mathbf{P}_e\right]_\times & \mathbf{0}_{3\times2} & \mathbf{0}_{3\times2} & \mathbf{0}_{3\times2} & ^G\mathbf{R}_B\mathbf{J}_4\left(\bar{\bar{\mathbf{\theta}}}_4\right) & \mathbf{0}_{3\times2} & \mathbf{0}_{3\times2} \\ \mathbf{I}_{3\times3} & -\left[^G\mathbf{R}_B, ^{5,B}\mathbf{P}_e\right]_\times & \mathbf{0}_{3\times2} & \mathbf{0}_{3\times2} & \mathbf{0}_{3\times2} & \mathbf{0}_{3\times2} & ^G\mathbf{R}_B\mathbf{J}_5\left(\bar{\bar{\mathbf{\theta}}}_5\right) & \mathbf{0}_{3\times2} \\ \mathbf{I}_{3\times3} & -\left[^G\mathbf{R}_B, ^{6,B}\mathbf{P}_e\right]_\times & \mathbf{0}_{3\times2} & \mathbf{0}_{3\times2} & \mathbf{0}_{3\times2} & \mathbf{0}_{3\times2} & \mathbf{0}_{3\times2} & ^G\mathbf{R}_B\mathbf{J}_6\left(\bar{\bar{\mathbf{\theta}}}_6\right) \end{bmatrix}$$

$$\dot{\bar{\bar{\mathbf{\theta}}}}_s = \begin{bmatrix} ^G\dot{\mathbf{P}}_B & ^G\mathbf{\Omega}_B & \dot{\theta}_{1,1} & \dot{\theta}_{1,2} & \dot{\theta}_{2,1} & \dot{\theta}_{2,2} & \dot{\theta}_{3,1} & \dot{\theta}_{3,2} & \dot{\theta}_{4,1} & \dot{\theta}_{4,2} & \dot{\theta}_{5,1} & \dot{\theta}_{5,2} & \dot{\theta}_{6,1} & \dot{\theta}_{6,2} \end{bmatrix}^T \tag{29}$$

The six-track robot is approximated as a differential drive robot, and the functional relationship between the control quantity which contains the left and right wheel speed $V_l$, $V_r$ with the body linear velocity $^G\dot{\mathbf{P}}_B$ and the body angular velocity $^G\mathbf{\Omega}_B$ is established:

$$\dot{X} = V_c cos\varphi_B$$

$$\dot{Y} = V_c \sin \varphi_B$$

$$^G\mathbf{\Omega}_B = \begin{bmatrix} 0 \\ 0 \\ \dot{\varphi}_B \end{bmatrix}, \quad V_c = \frac{v_r + v_l}{2}, \quad \dot{\varphi}_B = \frac{v_r - v_l}{L} \tag{30}$$

This model does not consider the slipping of the tracked robot on uneven terrain, whereas to prevent overdriving of the tracks during obstacle traversal, the model with slippage parameters proposed by [31] is used:

$$V_c = \frac{v_r(1 - \rho_r) + v_l(1 - \rho_l)}{2}$$

$$\dot{\varphi}_B = \frac{v_r(1 - \rho_r) - v_l(1 - \rho_l)}{2} \tag{31}$$

where

$$\frac{\rho_l}{\rho_r} = -sgn(v_r.v_l)\left|\frac{v_r}{v_l}\right|^n \tag{32}$$

where $n$ is a parameter concerning the track size, track tread, and track–ground contact properties and $sgn$ is a symbolic function. Consequently, the body linear velocity $^G\dot{\mathbf{P}}_B$ and the body angular velocity $^G\mathbf{\Omega}_B$ can be expressed as:

$$^G\dot{\mathbf{P}}_B = \mathbf{F}(\varphi_B, \rho_r, \rho_l)\begin{bmatrix} v_r \\ v_l \end{bmatrix} = \begin{bmatrix} \frac{(1-\rho_r)}{2}c_B & \frac{(1-\rho_l)}{2}c_B \\ \frac{(1-\rho_r)}{2}s_B & \frac{(1-\rho_l)}{2}s_B \\ 0 & 0 \end{bmatrix}\begin{bmatrix} v_r \\ v_l \end{bmatrix}$$

$$^{G}\mathbf{\Omega}_B = \mathbf{G}(\rho_r, \rho_l) \begin{bmatrix} v_r \\ v_l \end{bmatrix} = \begin{bmatrix} 0 & 0 \\ 0 & 0 \\ \frac{1-\rho_r}{2} & -\frac{1-\rho_l}{2} \end{bmatrix} \begin{bmatrix} v_r \\ v_l \end{bmatrix} \tag{33}$$

According to Equations (28) and (33), the final complete differential kinematic model of the six-track robot is given by:

$$^{G}\dot{\mathbf{P}}_e = \mathbf{J}\left(\bar{\bar{\mathbf{\theta}}}_s\right)\mathbf{u} \tag{34}$$

Among them,

$$^{G}\dot{\mathbf{P}}_e = \begin{bmatrix} ^{1,G}\dot{\mathbf{P}}_e & ^{2,G}\dot{\mathbf{P}}_e & ^{3,G}\dot{\mathbf{P}}_e & ^{4,G}\dot{\mathbf{P}}_e & ^{5,G}\dot{\mathbf{P}}_e & ^{6,G}\dot{\mathbf{P}}_e \end{bmatrix}^T$$

$$\mathbf{J}\left(\bar{\bar{\mathbf{\theta}}}_s\right) = \begin{bmatrix} \mathbf{F}(\varphi_B, \rho_r, \rho_l) - [^{1,G}\mathbf{P}_e - {}^{G}\mathbf{P}_B]_\times \mathbf{G}(\rho_r, \rho_l) & {}^{G}\mathbf{R}_B\mathbf{J}_1\left(\bar{\bar{\mathbf{\theta}}}_1\right) & \mathbf{0}_{3\times2} & \mathbf{0}_{3\times2} & \mathbf{0}_{3\times2} & \mathbf{0}_{3\times2} & \mathbf{0}_{3\times2} \\ \mathbf{F}(\varphi_B, \rho_r, \rho_l) - [^{2,G}\mathbf{P}_e - {}^{G}\mathbf{P}_B]_\times \mathbf{G}(\rho_r, \rho_l) & \mathbf{0}_{3\times2} & {}^{G}\mathbf{R}_B\mathbf{J}_2\left(\bar{\bar{\mathbf{\theta}}}_2\right) & \mathbf{0}_{3\times2} & \mathbf{0}_{3\times2} & \mathbf{0}_{3\times2} & \mathbf{0}_{3\times2} \\ \mathbf{F}(\varphi_B, \rho_r, \rho_l) - [^{3,G}\mathbf{P}_e - {}^{G}\mathbf{P}_B]_\times \mathbf{G}(\rho_r, \rho_l) & \mathbf{0}_{3\times2} & \mathbf{0}_{3\times2} & {}^{G}\mathbf{R}_B\mathbf{J}_3\left(\bar{\bar{\mathbf{\theta}}}_3\right) & \mathbf{0}_{3\times2} & \mathbf{0}_{3\times2} & \mathbf{0}_{3\times2} \\ \mathbf{F}(\varphi_B, \rho_r, \rho_l) - [^{4,G}\mathbf{P}_e - {}^{G}\mathbf{P}_B]_\times \mathbf{G}(\rho_r, \rho_l) & \mathbf{0}_{3\times2} & \mathbf{0}_{3\times2} & \mathbf{0}_{3\times2} & {}^{G}\mathbf{R}_B\mathbf{J}_4\left(\bar{\bar{\mathbf{\theta}}}_4\right) & \mathbf{0}_{3\times2} & \mathbf{0}_{3\times2} \\ \mathbf{F}(\varphi_B, \rho_r, \rho_l) - [^{5,G}\mathbf{P}_e - {}^{G}\mathbf{P}_B]_\times \mathbf{G}(\rho_r, \rho_l) & \mathbf{0}_{3\times2} & \mathbf{0}_{3\times2} & \mathbf{0}_{3\times2} & \mathbf{0}_{3\times2} & {}^{G}\mathbf{R}_B\mathbf{J}_5\left(\bar{\bar{\mathbf{\theta}}}_5\right) & \mathbf{0}_{3\times2} \\ \mathbf{F}(\varphi_B, \rho_r, \rho_l) - [^{6,G}\mathbf{P}_e - {}^{G}\mathbf{P}_B]_\times \mathbf{G}(\rho_r, \rho_l) & \mathbf{0}_{3\times2} & \mathbf{0}_{3\times2} & \mathbf{0}_{3\times2} & \mathbf{0}_{3\times2} & \mathbf{0}_{3\times2} & {}^{G}\mathbf{R}_B\mathbf{J}_6\left(\bar{\bar{\mathbf{\theta}}}_6\right) \end{bmatrix}$$

$$\mathbf{u} = \begin{bmatrix} v_r & v_l & \dot{\theta}_{1,1} & \dot{\theta}_{1,2} & \dot{\theta}_{2,1} & \dot{\theta}_{2,2} & \dot{\theta}_{3,1} & \dot{\theta}_{3,2} & \dot{\theta}_{4,1} & \dot{\theta}_{4,2} & \dot{\theta}_{5,1} & \dot{\theta}_{5,2} & \dot{\theta}_{6,1} & \dot{\theta}_{6,2} \end{bmatrix}^T \tag{35}$$

The differential kinematic model in Equation (34) defines the mapping relationship between the linear velocity $^{G}\dot{\mathbf{P}}_e$ of the six-track robot end points and the control quantity $\mathbf{u}$, which is used to implement the controller design of the tracked robot.

## 5. Autonomous Obstacle Traversal Controller for the Six-Track Robot

In this section, we will introduce the design method of autonomous obstacle traversal controller. The controller framework is shown in Figure 5, which mainly contains the control volume generation module, the inverse kinematic model module, and the feedback system. The control volume generation module overcomes the effect of suspension deformation on controller performance by separating the passive joints from the active joints. The inverse kinematic model module gives the method of calculation of the right pseudo-inverse of the Jacobi matrix and the singular position. The feedback system mainly consists of the positioning system, line displacement sensor feedback system, and motor feedback system.

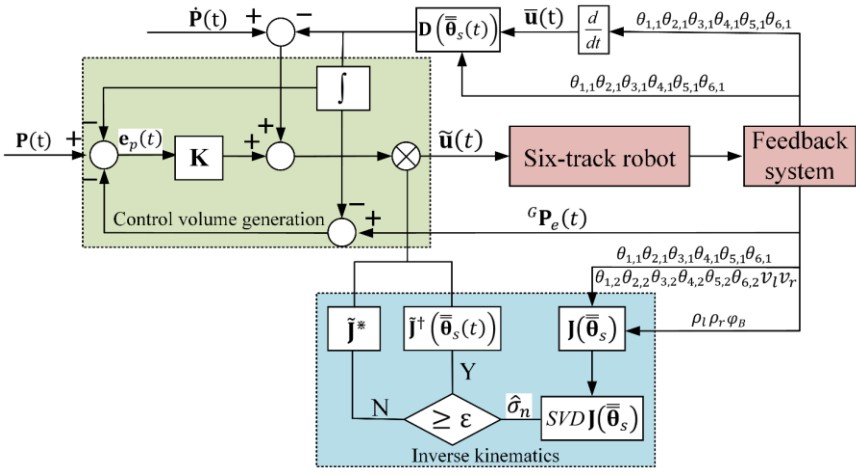

**Figure 5.** Autonomous obstacle traversal control algorithm framework.



### 5.1. Calculation of Control Volume

Let $\mathbf{P}(t)$ be the desired trajectory of the end point of the track that the robot needs to follow, and ${}^{G}\mathbf{P}_e(t)$ be the actual position of the end point of the track in the world coordinates system $\{C_G\}$, which can be expressed as:

$$
{}^{G}\mathbf{P}_e(t) = \left[ {}^{1,G}\mathbf{P}_e(t) \ {}^{2,G}\mathbf{P}_e(t) \ {}^{3,G}\mathbf{P}_e(t) \ {}^{4,G}\mathbf{P}_e(t) \ {}^{5,G}\mathbf{P}_e(t) \ {}^{6,G}\mathbf{P}_e(t) \right]^{T} \tag{36}
$$

The end point tracking error of the robot tracks can be expressed by the following equation:

$$
\mathbf{e}_p(t) = \mathbf{P}(t) - {}^{G}\mathbf{P}_e(t) \tag{37}
$$

The controller generates the control input $\mathbf{u}(t)$, according to the control error $\mathbf{e}_p(t)$, to enable the tracked robot to follow the desired trajectory. In order to make the tracking error converge quickly and steadily, the feedback control rate of the tracked robot is designed as follows. Since the joint angle $\theta_{i,1}$ in the established forward kinematic model is a passive rotational joint (as in Figure 1) which cannot be controlled during the obstacle traversal process, the differential kinematic equation in Equation (34) is rewritten in the form of the active joint separated from the passive joint, expressed as:

$$
{}^{G}\dot{\mathbf{P}}_e(t) = \tilde{\mathbf{J}}\left( \bar{\bar{\boldsymbol{\theta}}}_s(t) \right) \tilde{\mathbf{u}}(t) + \mathbf{D}\left( \bar{\bar{\boldsymbol{\theta}}}_s(t) \right) \bar{\mathbf{u}}(t)
$$

$$
\tilde{\mathbf{u}}(t) = \begin{bmatrix} v_r(t) \\ \dot{v}_r(t) \\ \dot{\theta}_{1,2}(t) \\ \dot{\theta}_{2,2}(t) \\ \dot{\theta}_{3,2}(t) \\ \dot{\theta}_{4,2}(t) \\ \dot{\theta}_{5,2}(t) \\ \dot{\theta}_{6,2}(t) \end{bmatrix}, \bar{\mathbf{u}}(t) = \begin{bmatrix} \dot{\theta}_{1,1}(t) \\ \dot{\theta}_{2,1}(t) \\ \dot{\theta}_{3,1}(t) \\ \dot{\theta}_{4,1}(t) \\ \dot{\theta}_{5,1}(t) \\ \dot{\theta}_{6,1}(t) \end{bmatrix} \tag{38}
$$

Among them,

$$
\tilde{\mathbf{J}}\left( \bar{\bar{\boldsymbol{\theta}}}_s(t) \right) = [J_{s1} \ J_{s2} \ J_{s4} \ J_{s6} \ J_{s8} \ J_{s10} \ J_{s12} \ J_{s14}] \in \mathbb{R}^{6 \times 8}
$$

$$
\mathbf{D}\left( \bar{\bar{\boldsymbol{\theta}}}_s(t) \right) = [J_{s3} \ J_{s5} \ J_{s7} \ J_{s9} \ J_{s11} \ J_{s13}] \in \mathbb{R}^{6 \times 6} \tag{39}
$$

where $J_{si}$ is the column vector of the Jacobi matrix $\mathbf{J}\left( \bar{\bar{\boldsymbol{\theta}}}_s \right)$ at moment $t$. Equation (38) shows that this controller can directly calculate the control input $\tilde{\mathbf{u}}(t)$ containing only the active joints according to the desired track end point line speed, so the effect of the passive joints is independent of the control rate. Thus, differentiating Equation (37) with respect to time yields:

$$
\dot{\mathbf{e}}_p(t) = \dot{\mathbf{P}}(t) - \tilde{\mathbf{J}}\left( \bar{\bar{\boldsymbol{\theta}}}_s(t) \right) \tilde{\mathbf{u}}(t) - \mathbf{D}\left( \bar{\bar{\boldsymbol{\theta}}}_s(t) \right) \bar{\mathbf{u}}(t) \tag{40}
$$

In order to make the control error $\mathbf{e}_p(t)$, as established in Equation (37), converge asymptotically and steadily to zero, it is linearized as follows:

$$
\dot{\mathbf{e}}_p(t) - \mathbf{K}\mathbf{e}_p(t) = 0 \tag{41}
$$

where $\mathbf{K}$ is the gain matrix, which is a positive definite matrix, when the system can be asymptotically stable. According to the error differential model, described in Equation (38),

as well as the linear system in Equation (41), the active joint control input $\tilde{\mathbf{u}}(t)$ can be obtained using the following equation:

$$\tilde{\mathbf{u}}(t) = \tilde{\mathbf{J}}^{\dagger}\left(\bar{\bar{\boldsymbol{\theta}}}_s(t)\right)\left(\dot{\mathbf{P}}(t) - \mathbf{D}\left(\bar{\bar{\boldsymbol{\theta}}}_s(t)\right)\bar{\mathbf{u}}(t) + \mathbf{K}\mathbf{e}_p(t)\right) \tag{42}$$

where $\tilde{\mathbf{J}}^{\dagger}\left(\bar{\bar{\boldsymbol{\theta}}}_s(t)\right)$ is the right pseudo-inverse matrix of the Jacobi matrix $\tilde{\mathbf{J}}\left(\bar{\bar{\boldsymbol{\theta}}}_s(t)\right)$ at time $t$. The convergence rate of the system error depends on gain matrix $\mathbf{K} \in \mathbb{R}^{18 \times 18}$. The larger the eigenvalue of the gain matrix, the faster the convergence rate of the system error. Therefore, the active joint control input $\tilde{\mathbf{u}}(t)$, calculated in Equation (42), leads to the following operational positional space dynamics:

$$\dot{\mathbf{e}}_p(t) = \dot{\mathbf{P}}(t) - \mathbf{D}\left(\bar{\bar{\boldsymbol{\theta}}}_s(t)\right)\bar{\mathbf{u}}(t) - \tilde{\mathbf{J}}\left(\bar{\bar{\boldsymbol{\theta}}}_s(t)\right)\tilde{\mathbf{J}}^{\dagger}\left(\bar{\bar{\boldsymbol{\theta}}}_s(t)\right)\left(\dot{\mathbf{P}}(t) - \mathbf{D}\left(\bar{\bar{\boldsymbol{\theta}}}_s(t)\right)\bar{\mathbf{u}}(t) + \mathbf{K}\mathbf{e}_p(t)\right) \tag{43}$$

where $\tilde{\mathbf{J}}\left(\bar{\bar{\boldsymbol{\theta}}}_s(t)\right)\tilde{\mathbf{J}}^{\dagger}\left(\bar{\bar{\boldsymbol{\theta}}}_s(t)\right) = \mathbf{I}$ is a unit matrix. It can be easily proven that the computed control inputs can ensure stable convergence of the system. Ultimately, the six flipper ends of the tracked robot can follow the specified desired trajectory. The control input $\tilde{\mathbf{u}}(t)$ is computed by Equation (42), making the system error converge asymptotically, whereas the asymptotic stability of the system under this control input is demonstrated by the virtual simulation experiments in Section 6.

### 5.2. Kinematic Singularities

The inversion of the Jacobian can represent a serious inconvenience not only at a singularity but also in the neighborhood of a singularity. This situation is characterized by the loss of some spatial freedom of operation of the robot, implying that low desired velocities in corresponding directions will lead to extremely high joint velocities. This behavior is particularly problematic for inverse kinematics algorithms, as introduced in Section 5.1. Therefore, the singularity avoidance treatment of Jacobi matrices must be performed. An alternative solution, overcoming the problem of inverting differential kinematics in the neighborhood of a singularity, is provided by the so-called damped least squares (DLS) inverse, for which a velocity damping term is introduced, making a compromise between tracking accuracy and joint speed, i.e., satisfying:

$$\tilde{\mathbf{J}}^{*} = \tilde{\mathbf{J}}^{T}\left(\tilde{\mathbf{J}}\tilde{\mathbf{J}}^{T} + \eta^2 \mathbf{I}\right)^{-1} \tag{44}$$

where $\eta$ is a damping factor that renders the inversion better conditioned from a numerical viewpoint. In order to find the solution $\tilde{\mathbf{u}}(t)$, Equation (38) is rewritten as the following linear equation:

$${}^{G}\dot{\mathbf{P}}_e(t) - \mathbf{D}\left(\bar{\bar{\boldsymbol{\theta}}}_s(t)\right)\bar{\mathbf{u}}(t) = \tilde{\mathbf{J}}\left(\bar{\bar{\boldsymbol{\theta}}}_s(t)\right)\tilde{\mathbf{u}}(t) \tag{45}$$

It can be shown that such a solution can be obtained by reformulating the problem in terms of the cost functional minimization:

$$g\left(\tilde{\mathbf{u}}(t)\right) = \frac{1}{2}\left\|{}^{G}\dot{\mathbf{P}}_e(t) - \mathbf{D}\left(\bar{\bar{\boldsymbol{\theta}}}_s(t)\right)\bar{\mathbf{u}}(t) - \tilde{\mathbf{J}}\left(\bar{\bar{\boldsymbol{\theta}}}_s(t)\right)\tilde{\mathbf{u}}(t)\right\|^2 + \frac{1}{2}\eta^2\left\|\tilde{\mathbf{u}}(t)\right\|^2 \tag{46}$$

The introduction of the first term allows a finite inversion error to be tolerated, with the advantage of norm-bounded velocities. The factor $\eta$ establishes the relative weight between the two objectives, while there are techniques for selecting optimal values for the damping factor. However, this approach compromises the accuracy in all directions at the end of the robot. Specifically, low values of $\eta$ can lead to the accurate solution of $\tilde{\mathbf{J}}^{*}$, but the robustness is reduced around singular values, whereas high values of $\eta$ lead

to a reduction in tracking accuracy. This indicates that it is difficult to meet the robot performance requirements while $\eta$ remains constant throughout the operating position space. It is common practice to dynamically adjust $\eta$ by choosing a threshold that can represent the singular region. The results show that the damping factor can be adaptively adjusted according to the following equation:

$$
\eta^2 = \begin{cases} 0, & \hat{\sigma}_n \geq \varepsilon \\ \left(1 - \left(\dfrac{\hat{\sigma}_n}{\varepsilon}\right)^2\right)\lambda_m^2, & other \end{cases} \tag{47}
$$

where $\hat{\sigma}_n$ is the estimate of the minimum singular value, $\varepsilon$ is the set threshold for determining whether the robot is singular or not, and $\lambda_m$ is the maximum damping value of the singular region. For more details on the dynamic adjustment of the damping factor $\eta$, a detailed analysis can be found in [32].

## 6. Experiments

### 6.1. Virtual Simulation Experiments

Two scenarios are designed in the virtual simulation experiment: (1) S-curve terrain scenario, designed to verify the performance of the autonomous obstacle traversal controller in field rescue missions, including the same tracking capabilities as the 2D trajectory tracking controller and generation of appropriate joint commands, ensuring that the six-track robot can approach obstacles correctly and successfully navigate through complex environments; (2) single-sided step obstacle scenario, designed to verify the independent control capability of each flipper during the execution of the task. In the simulation experiments, the terrain and robot models are built in the virtual robot simulation platform CoppeliaSim [33], which has an open dynamics engine to realistically simulate the interaction between the robot and the environment.

### 6.1.1. S-Curve Scenario Experiment Phrasing

The S-curved scenario contains a step obstacle 0.35 m high, 6 m long, and 8 m wide as well as a slope obstacle 6 m long and 4 m wide with a slope of $30°$. The collision-free desired trajectory of the six-track robot in the virtual simulation platform is planned manually. On flat terrain, the planned desired path is located at a specific distance above the ground determined by the physical size of the tracks in order to restore the state of the six-track robot moving at high speed on wheels. On slopes and step terrain, the planned desired path is located on the obstacle surface. The robot is placed at an initial position 2 m offset from the planned path with an initial heading angle of 30 degrees. The controller will generate appropriate control commands, in order to guide the robot to appropriately approach the sloping terrain and obstacle terrain.

The performance of the controller is evaluated by analyzing the system asymptotic stability and trajectory tracking accuracy, while the track is configured so that there is no relative sliding against the terrain; that is, the slip parameter in Equation (31) is zero. The actual trajectories of the tracked robot center of mass and the six flipper end points are illustrated in Figure 6 using different colors. Figure 7a–c show the trajectory error norm of front, middle, and rear flipper end points, respectively, and Figure 7d shows the center of mass trajectory error norm. The initial large norm error is due to the robot being placed at the initial position 2 m offset from the planned path, while the convergence speed of the trajectory error norm depends on the eigenvalues of the gain matrix $\mathbf{K}$. When the gain matrix $\mathbf{K} \in \mathbb{R}^{18 \times 18}$ in Equation (42) is $\mathbf{K} = diag\{\lambda_1, \ldots, \lambda_{18}\}$, where $\lambda_k = 0.4, k \in 1, \ldots, 18$, the convergence rate of the trajectory error norm of the six flippers is about 0.998, and the asymptotic stability of the system is reflected.

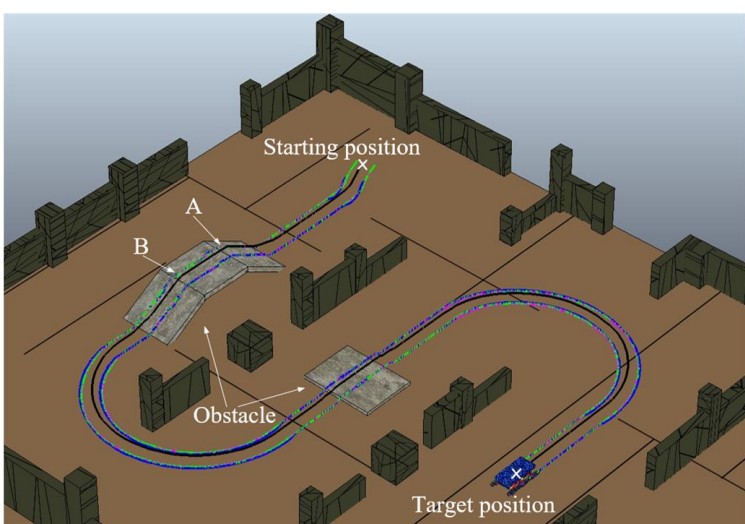

**Figure 6.** S-curve scenario experiment with slopes and short step obstacles. Point A indicates that the robot is in a raised pose at the climbing end stage. Point B indicates that the rapid drop of the robot under the effect of gravity after the robot passes through the flat slope.

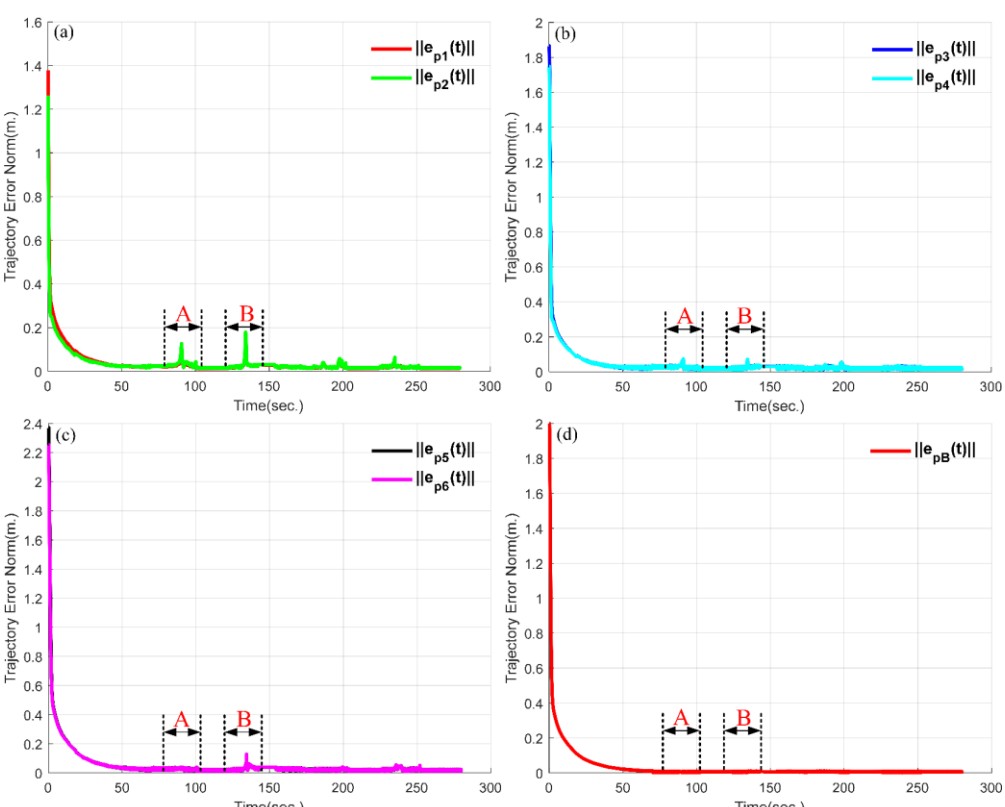

**Figure 7.** Asymptotic stability analysis and tracking accuracy analysis in S-curve terrain scenario. Subfigure (**a**) indicates the error norms of the front flipper. Subfigure (**b**) indicates the error norms of the middle flipper. Subfigure (**c**) indicates the error norms of the rear flipper. Subfigure (**d**) indicates the error norm of the center of mass. A indicates that the increase of error norm is due to the robot in a raised pose at the climbing end stage. B indicates that the increase of error norm is due to the rapid drop of the robot under the effect of gravity.

The tracking accuracy of the controller is evaluated according to the following criteria: (1) the trajectory error norm $\|\mathbf{e}_{P_i}\|$ of the six flippers and (2) the trajectory error norm $\|\mathbf{e}_{P_B}\|$ of the robot's center of mass. It is noted that within the timeframe $\Delta t = [85, 95]$, the front

flipper error norm $\|\mathbf{e}_{P_1}\|$ and error norm $\|\mathbf{e}_{P_2}\|$ increase rapidly from 0.02 m to 0.1 m, while the rest of the flipper error norm is stable at about 0.02 m. This situation occurs because the robot is in a raised pose at the climbing end stage, as shown at point A in Figure 5. By following the desired path, it exceeds the working space of the front flipper, so the desired trajectory cannot be tracked effectively, while the influence of this attitude on the middle and rear tracks is smaller. Similarly, within the timeframe $\Delta t = [135, 145]$, the error norm $\|\mathbf{e}_{P_1}\|$ and error norm $\|\mathbf{e}_{P_2}\|$ of the front flipper increases from 0.013 m to 0.19 m, respectively, which is caused by the rapid drop of the robot under the effect of gravity after the robot passes through the flat slope, as shown at point B of Figure 5. It is worth noting that the effect of this drop on the error norm of the rear flipper was greater than that of the middle flipper, which is consistent with what is generally known. During the timeframe $\Delta t = [120, 130]$, the robot is in the stage of traversing the step obstacle, while the error norm $\|\mathbf{e}_{P_1}\|$ and error norm $\|\mathbf{e}_{P_2}\|$ vary slightly between 0.02 m and 0.05 m, respectively, which is mainly caused by the abrupt raising of the track main wheel.

On the horizontal terrain, the six-track robot can move at high speed using the wheel, while the tracking error of the flipper trajectory is constant at 0.02 m, which shows high tracking accuracy. Furthermore, under the effect of gravity, the collision causes a small deviation of the robot's center of mass, as shown in Figure 7d. However, the controller can generate the appropriate control commands to rapidly stabilize the center of mass trajectory error norm and avoid the trajectory deviation and heading oscillation. The experiment proves that the autonomous obstacle traversal controller enables synchronous and stable convergence of flipper error and track traction within a unified control framework while it maintains a high tracking accuracy.

### 6.1.2. Single-Sided Step Scenario Experiment

A single-sided step scenario consists of four steps, with each step having a height of 0.2 m, a length of 2 m, and a width of 2 m. The robot was placed at an initial position of 5 m from the step, which ensured that the robot could only pass the step on the left side, while the right side always walked on the ground, as it was meant to provide sufficient friction and prevent the robot from tipping over.

The actual trajectories of the six flippers and the center of mass of the tracked robot are shown in Figure 8, whereas the trajectory error norm of the six flippers during the task is reported in Figure 9a–c. The trajectory error norm of the robot center of mass is illustrated in Figure 9d.

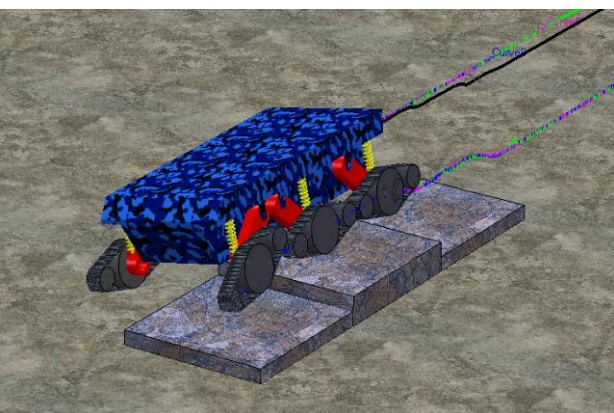

**Figure 8.** Six-track robot traverses a single-sided step scenario, which consists of four steps with height of 0.2 m, where only one side of the tracked robot is allowed to pass.

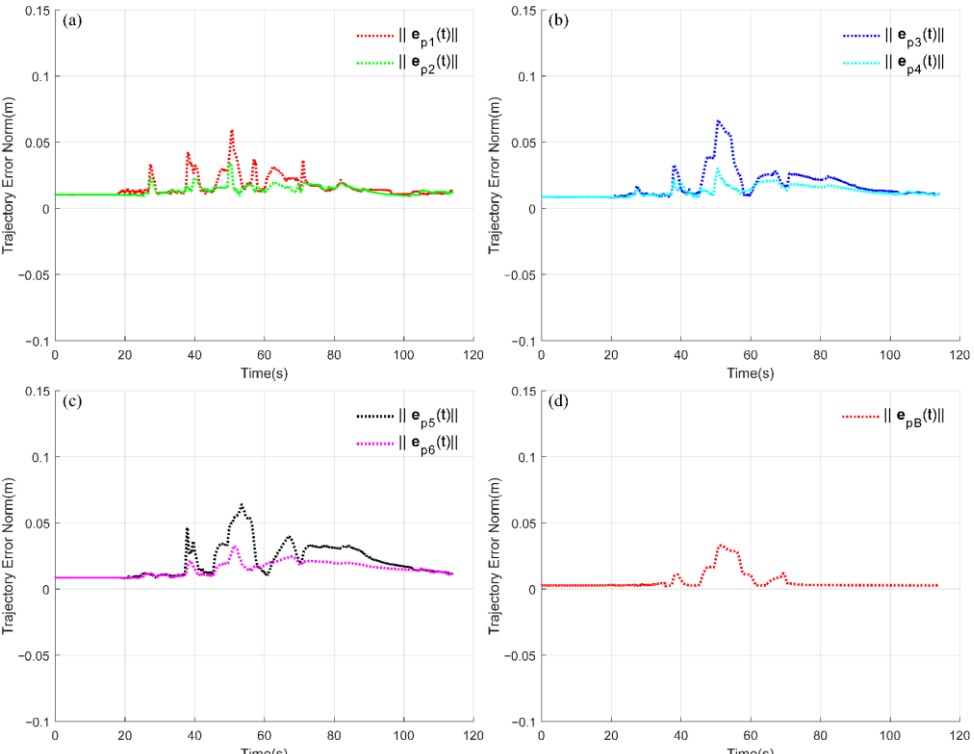

**Figure 9.** Tracking accuracy analysis in single-sided step scenario. Subfigure (**a**) indicates the error norms of the front flipper. Subfigure (**b**) indicates the error norms of the middle flipper. Subfigure (**c**) indicates the error norms of the rear flipper. Subfigure (**d**) indicates the error norm of the center of mass.

According to the conclusion, the left trajectory error norm of the tracked robot is always larger than the right trajectory error norm, which is caused by the undulation of the left terrain. Among them, the trajectory error of the flipper increases slightly at t = 28 s, which is mainly caused by the main wheel climbing. Around the timestamps t = 40 s and t = 68 s, the increase of the robot body roll angle causes the increase of the center of mass trajectory error norm, which can recover quickly under the action of the controller, while the error norm of the flipper increases to about 0.04 m. During the traversal of the second step, the further increase of the robot's roll angle causes an increase in the mass trajectory error norm to 0.034 m, while the flipper trajectory error norm increases to 0.063 m. During the obstacle traversal, the average trajectory error of the flipper was 0.029 m, the time of the obstacle crossing was 68 s, and the maximum roll angle of the robot was 21°. The experiment proves that the designed controller can independently control the six flippers according to the planned trajectory and realize high-precision autonomous obstacle traversal.

### 6.2. Physical Prototype Experiment

In order to obtain the performance of the autonomous obstacle traversal controller in comparison to the manual remote control and the existing controller under uniform index, traversal of circular experimental fields with short steps, trenches, and slopes is performed. In the case of the circular experimental fields in the RVIZ (Figure 10), the environmental data are acquired by an on-board 32-line LiDAR and replicated using SLAM technology. In the process of obstacle traversal in the field terrain, the suspension is deformed to different degrees due to uneven force, and the influence of this deformation on the designed controller is mainly reflected in the passive joint angle and angular velocity, so it is necessary to establish the mapping relationship between the suspension deformation displacement and the passive joint. We measure the deformation of the suspension in real

time through the line displacement sensor, which is installed at the position parallel to the suspension, and the mapping between the suspension displacement and the angular and angular velocities of the passive joints is completed through geometric relations. Then, the passive joint is separated from the active joint in the control algorithm so that the influence of the suspension deformation on the robot motion and control performance can be eliminated.

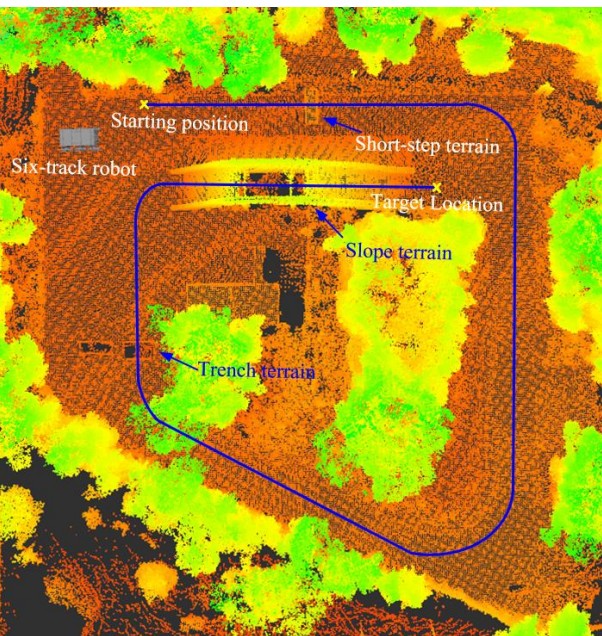

**Figure 10.** Circular experimental fields with short step, trench, and slope terrain. The blue line is the planned path that the tracked robot must follow.

The position information of the complete baseline path is obtained via GPS fixed-point sampling and generated via interpolation, which will be used as the desired path for the manual remote control and the prior art obstacle traversal controller. In addition, this path position information is combined with a forward kinematic model of the tracked robot, while the terrain parameters of the circular experimental field acquired by the LIDAR are also considered in order to generate the desired path that can be tracked by the autonomous obstacle traversal controller. The prior art obstacle traversal controller was described in [34], in which the center of mass controller and the flipper position controller are independent of each other. The hardware of the six-tracked robot includes remote control, a wireless transceiver module, an industrial computer, a drive motor, a gait motor, a battery, a line displacement sensor, LIDAR and GPS/IMU, etc. The remote control can send different mode commands, and the control algorithm is deployed in the Ubuntu system of the industrial computer. The hardware block diagram is shown in Figure 11.

Snapshots of the six-track robot traversing the trench terrain of the circular experimental field are shown in Figure 12, while the flipper error norm and the center of mass error norm are illustrated in Figure 13. The width of the trench terrain is 0.8 m, the length is 3 m, and the height is 0.4 m. The tracked robot has a load of 500 kg and crosses the trench at 0.16 m/s. Observations showed that during the trench traversal, there were two relatively significant increases of the flipper error norm. The first one occurs in the timeframe $\Delta t = [14, 16]$, the front flipper error norm increases from 0.04 m to 0.11 m as the front flipper nears the other end of the trench, and the robot broke its equilibrium under the influence of gravity, as shown in Figure 12c. The second increase occurs in the timeframe $\Delta t = [20, 22]$, when there is a slight increase in error norm of the front flipper and rear flipper, which occurs when the robot's rear flipper is removed from the trench, as shown in Figure 12e. However, these two cases have less influence on the middle flipper error norm,

which is consistent with what is known. Additionally, in the whole trench traversal, the center of mass error norm is stable at about 0.04 m.

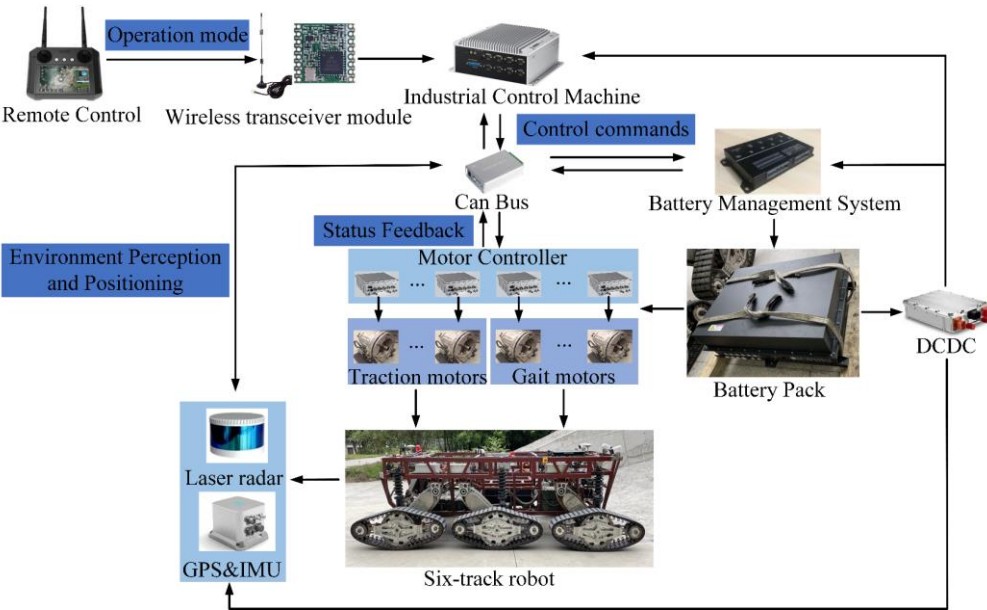

**Figure 11.** Six-track robot hardware block diagram.

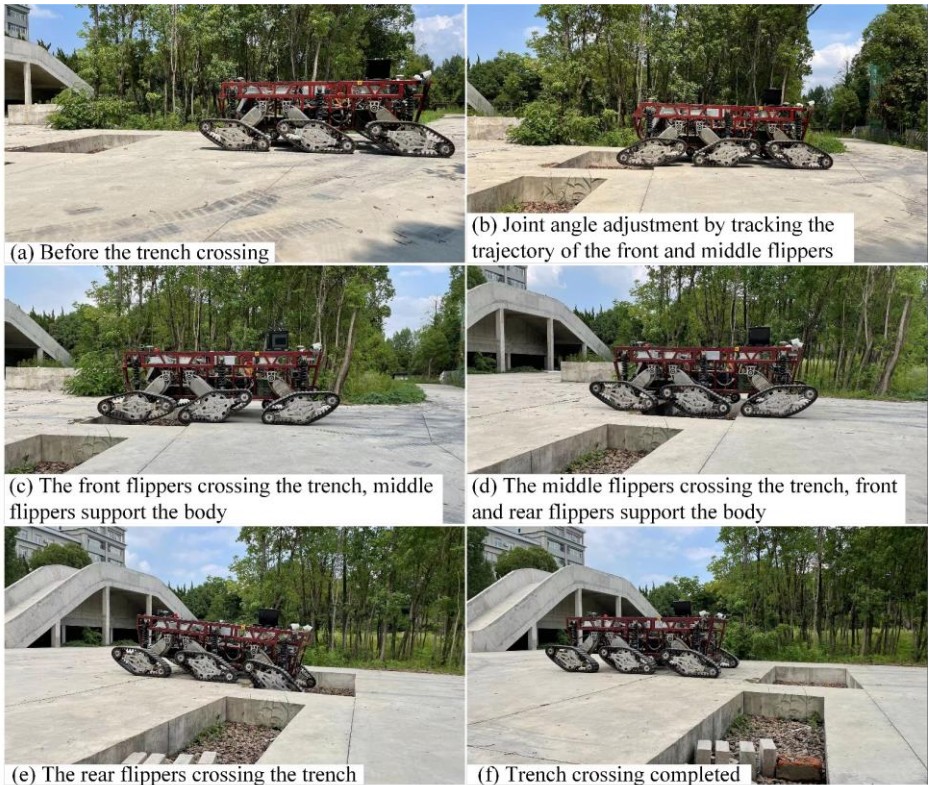

**Figure 12.** Snapshot of the autonomous controller traversing the trench terrain.

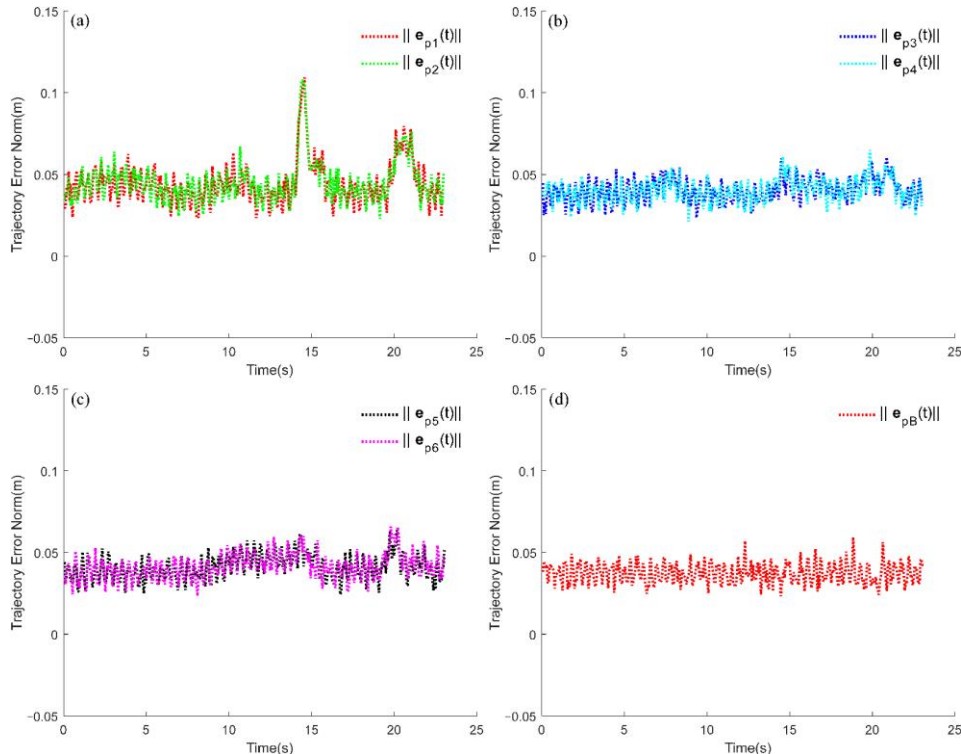

**Figure 13.** Trajectory error norm of tracked robot through trench terrain. Subfigure (**a**) indicates the error norms of the front flipper. Subfigure (**b**) indicates the error norms of the middle flipper. Subfigure (**c**) indicates the error norms of the rear flipper. Subfigure (**d**) indicates the error norm of the center of mass.

Snapshots of the six-track robot traversing the short step terrain are shown in Figure 14, whereas the flipper error norm and the center of mass error norm are shown in Figure 15. The width of the short step terrain is 4 m, the length is 1.2 m, and the height is 0.4 m. The slope of the terrain is $30°$, and its length is 5 m. The different forces on the front and rear suspensions caused by the pose change and the track deformation, lead to a large jitter of the flipper error norm. During the time spans $\Delta t = [2, 4]$ and $\Delta t = [14, 16]$, the error norms of the front flipper and middle flipper increase slightly due to the main track being over the obstacle, whereas the rear flipper remains almost unaffected. The rapid increase in the flipper error norm within the time frames $\Delta t = [7, 9]$ and $\Delta t = [10, 14]$ is due to the robot body being in a raised state (as in Figure 14c) and the front flipper being in a supported state after falling (as in Figure 14d), leading the desired trajectory out of the flipper working space. It is worth noting that during the horizontal plane terrain, the six-track robot can move at high speed using its wheels; during the slope-climbing phase, as shown in Figure 16, the controller significantly reduces the robot's driving speed to reduce the tracking error of the flippers, reflecting the advantage of simultaneous convergence of all tracking errors under a unified control framework.

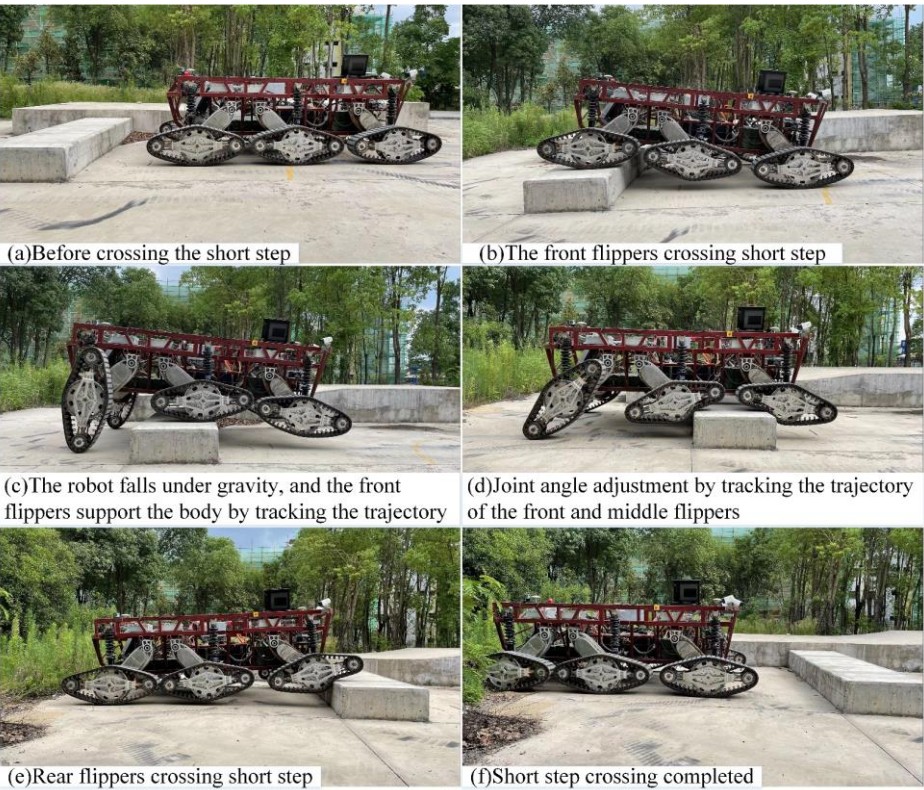

**Figure 14.** Snapshot of short step traversal.

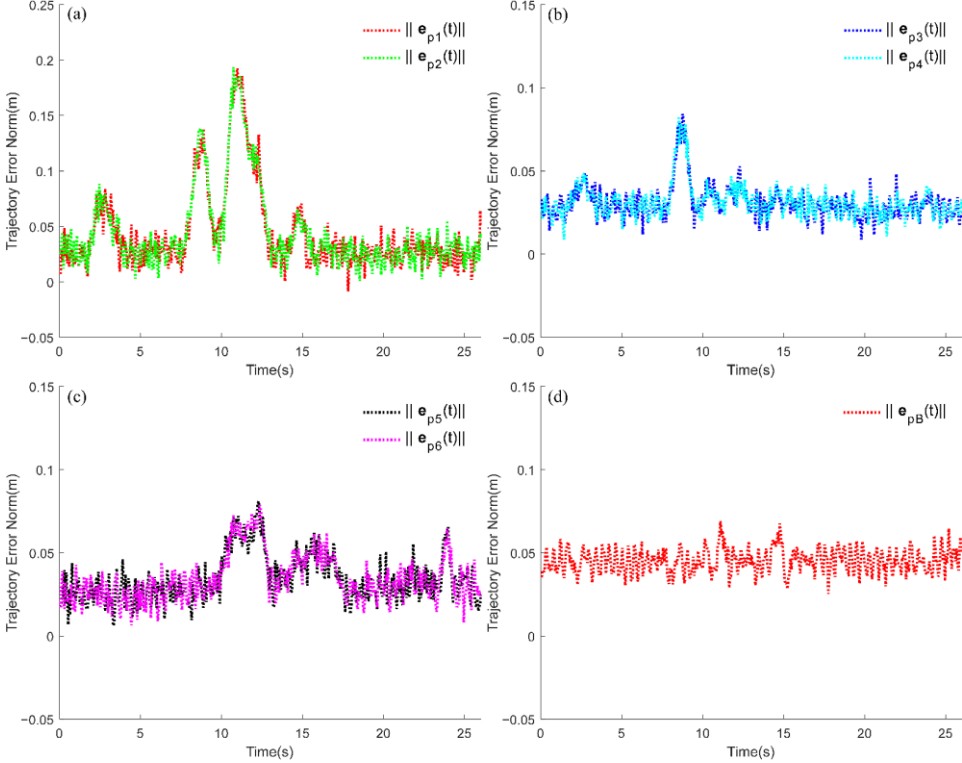

**Figure 15.** Trajectory error norm of tracked robots through short step. Subfigure (**a**) indicates the error norms of the front flipper. Subfigure (**b**) indicates the error norms of the middle flipper. Subfigure (**c**) indicates the error norms of the rear flipper. Subfigure (**d**) indicates the error norm of the center of mass.

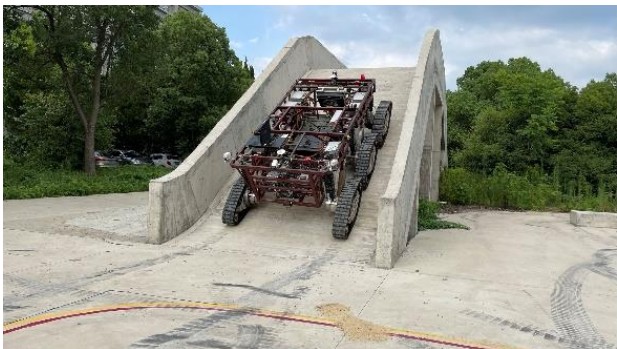

**Figure 16.** Snapshot of the autonomous controller traversing the slope terrain.

Table 3 shows the performance comparison results of the three controllers: (1) the mean error norm of the center of mass $\mathbf{e}_B(t)$, (2) the obstacle traversal time $t$, (3) the slipping norm $\Delta\sigma$, and (4) the maximum response, including the maximum linear and angular velocities. The wide body of the six-track robot complicates its turn maneuver, especially in the turning and obstacle traversal phases. In addition, the inclined path in the circular experimental fields increases the adjustment process. The autonomous obstacle traversal controller considers the slip effect between the track and the terrain, while it can follow the trajectory with high precision, reducing the average error norm by 40.7% and 13.5%, while the maximum slip norm is reduced by 34.6% and 19.9% compared to manual remote control and prior art controller, respectively. Furthermore, the planned path requires the six-track robot to move at high speed on wheels on flat terrain, while the synchronous convergence of flipper tracking error and mass center tracking error reduces the influence of traction speed on the obstacle traversal, thus limiting obstacle traversal time by 21.3% and 9.3%, respectively. The performance of the autonomous obstacle traversal controller is lower, in terms of response speed. In fact, the control rate established in Equation (42) determines the response speed of the controller with respect to the error, while higher tracking error leads to the increase of the response speed. The relationship between the generalized speed of the tracked robot and the slipping norm is given in Figure 17. The influence of linear speed on the slip rate along straight road section is mainly exhibited in the stage of rapid speed decline of the robot, which is mainly caused by the braking force. During a turning movement of the robot, the slip between the track and the terrain will further increase. However, Figure 17 shows that only when the angular velocity reaches 0.25 rad/s will the slip be affected, which proves the effectiveness of introducing slip compensation into the controller.

**Table 3.** The performance comparison results of the three controllers that traversed the circular experimental field.

| Control Method | Mean $\|\mathbf{e}_B(t)\|$ (m) | Max $\Delta\sigma$ (m) | $v_{max}$ (m/s) | $\omega_{max}$ (rad/s) | Time (s) |
|---|---|---|---|---|---|
| Manual remote control | 0.194 | 0.228 | 0.824 | 0.723 | 197 |
| Controller in the prior art | 0.133 | 0.186 | 0.682 | 0.515 | 171 |
| Autonomous obstacle traversal controller | 0.115 | 0.149 | 0.627 | 0.491 | 155 |

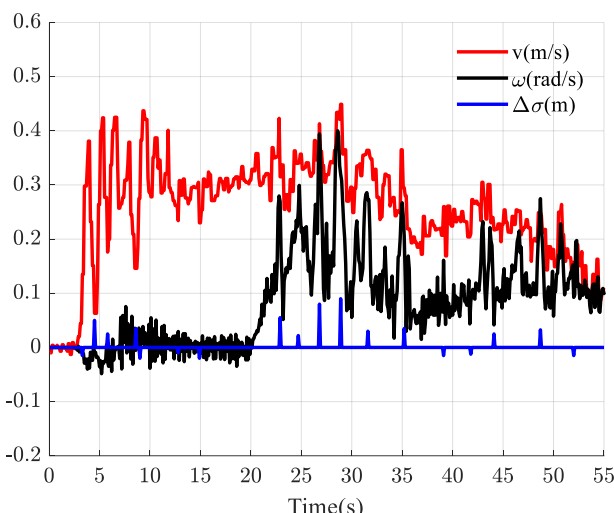

**Figure 17.** The relationship between the generalized speed of the tracked robot and the slipping norm.

## 7. Conclusions

In this paper, a new six-track robot with wheeled, legged, and tracked advantages is developed. Compared to the tracked robot with active flippers, the six-track robot has high-speed driving ability, large load capacity, and higher mobility, and it can better adapt to flat, soft, and obstructed terrains. A design method for a distributed autonomous obstacle-crossing controller based on differential kinematics is proposed. A forward kinematics model and a differential kinematics model of the six-track robot is established, and a unified control scheme including all degrees of freedom of the robot is provided. The control scheme integrates the slip-steering model to account for both the track–terrain interaction and the flipper's terrain adaptability. In addition, feedback control rates of the linear system and optimal solution for singular position are derived.

Virtual simulation experiments and physical prototype experiments involving various obstacle scenarios are performed. (1) The virtual simulation experiment contains an S-curve terrain scenario and a single-sided step terrain scenario. In the S-curve scenario experiment, the linear system can converge rapidly and steadily under the designed feedback control rate, while the convergence rate is about 1. At the climbing end stage, when the robot in a raised pose, the front flippers' error norms increase rapidly from 0.02 m to 0.1 m, while the rest of the flippers' error norms are stable at about 0.02 m. At the robot rapid drop stage, under the effect of gravity, the error norm of the front flippers increased from 0.013 m to 0.19 m. During the rest of the phase, the tracking errors of the flippers' trajectories are constant at 0.02 m. Therefore, the controller has a high tracking accuracy except for the uncontrollable obstacle traversal process. In the single-sided step terrain scenario, the maximum error norms of the flipper are 0.04 and 0.063 at the first and second stage, respectively, and the average trajectory error of the flippers is about 0.029 m, verifying that the controller can achieve high-precision obstacle traversal and generate appropriate joint and speed commands as well as the independent control capability of each flipper. (2) The physical prototype experiment contains terrain scenarios such as short step, trench, and slope. In the whole trench traversal, the center of mass error norm was stable at about 0.04 m. Compared to the manual remote controller and the prior art controller, the average error norm of the center of mass is reduced by 40.7% and 13.5%, respectively; the maximum slip norm is reduced by 34.6% and 19.9%, respectively; and the obstacle crossing time is reduced by 21.3% and 9.3%, respectively. The results show that, the controller can increase the terrain adaptability of the tracked robot by realizing more advanced actions so as to achieve efficient and accurate obstacle-crossing. In addition, the controller effectively reduces the slippage effect between the track and the terrain. It is worth noting that when the flipper tracking error is large, the control system gives a lower traction speed command, which enables the robot mass center trajectory tracking error and the flipper trajectory

tracking error to converge synchronously, validating the advantages of the central controller and the flipper controller operating in a unified framework.

In the future, research on the pose control of tracked robots will be carried out and distributed dynamics will be established. This research will better match the movement characteristics of the six-track robot in the field terrain and improve the robot's mobility and stability.

**Author Contributions:** R.B. was in charge of the establishment of the model and the whole trial; R.B. and R.N. wrote the manuscript; J.W. and Z.X. assisted with experimentation. All authors have read and agreed to the published version of the manuscript.

**Funding:** This work was supported by National Key Research and Development Program of China (2020AAA0108103), Supported by the Independent Project of the Institute of Robotics and Intelligent Manufacturing Innovation of the Chinese Academy of Sciences (C2021002) and President's Foundation of Hefei Institutes of Physical Science, Chinese Academy of Sciences (YZJJZX202013).

**Data Availability Statement:** Not applicable.

**Conflicts of Interest:** The authors declare no conflict of interest.

## Nomenclature

| | |
|---|---|
| $\{C_G\} = \{X_G, Y_G, Z_G\}$ | World coordinate system |
| $\{C_0\} = \{X_0, Y_0, Z_0\}$ | The base link coordinate system which fixed on the robot center of mass |
| $\{C_B\} = \{X_B, Y_B, Z_B\}$ | The body coordinate system which fixed on the robot center of mass |
| $\left\{{}^iC_j\right\} = \left\{X_{i,j}, Y_{i,j}, Z_{i,j}\right\}$ | The $j$th coordinate system which fixed on the rotational joint of the $i$th leg of the robot, with $i \in \{1,2,3,4,5,6\}$ and $j \in \{1,2,3\}$ |
| $\theta_{i,j}$ | The $j$th joint angle of the $i$th leg of robot, with $i \in \{1,2,3,4,5,6\}$ and $j \in \{1,2,3\}$ |
| $\theta_B$ | Heading angle of robot |
| ${}^{i,j}\mathbf{P}_{j+1}$ | The position vector of the coordinate system $\left\{C_{j+1}\right\}$ with respect to the coordinate system $\left\{C_j\right\}$ in the $i$th leg, with $i \in \{1,2,3,4,5,6\}$ and $j \in \{1,2,3\}$ |
| ${}^G\mathbf{P}_B$ | The position vector of the body coordinate system $\{C_B\}$ with respect to world coordinate system $\{C_G\}$ |
| ${}^G\mathbf{\Omega}_B$ | The angular velocity of the body coordinate system $\{C_B\}$ with respect to the world coordinate system $\{C_G\}$ |
| ${}^{i,j}\boldsymbol{\omega}_{j,j+1}$ | The angular velocity of the coordinate system $\left\{{}^iC_{j+1}\right\}$ relative to the coordinate system $\left\{{}^iC_j\right\}$ in the $i$-th leg kinematic chain, expressed in the coordinate system $\left\{{}^iC_j\right\}$, with $i \in \{1,2,3,4,5,6\}$ and $j \in \{1,2,3\}$ |
| $v_l, v_r$ | Traction speed of the left track and traction speed of the right track |
| ${}^{i,G}\mathbf{P}_e$ | The position vector of the flipper end point in the $i$th leg kinematic chain with respect to the world coordinate system $\{C_G\}$, with $i \in \{1,2,3,4,5,6\}$ |
| ${}^{i,B}\mathbf{P}_e$ | The position vector of the flipper end point in the $i$th leg kinematic chain with respect to the body coordinate system $\{C_B\}$, with $i \in \{1,2,3,4,5,6\}$ |
| ${}^{i,G}\dot{\mathbf{P}}_e$ | The linear velocity of the flipper end point in the $i$th leg kinematic chain with respect to the world coordinate system $\{C_G\}$, with $i \in \{1,2,3,4,5,6\}$ |
| ${}^{i,B}\dot{\mathbf{P}}_e$ | The linear velocity of flipper end point in the $i$th leg kinematic chain with respect to the body coordinate system $\{C_B\}$, with $i \in \{1,2,3,4,5,6\}$ |
| $\bar{\bar{\theta}}_s$ | Configuration of joint angles for six-track robots |
| $\mathbf{u}$ | Control command vector |
| $\bar{\bar{\theta}}_i$ | Configuration of joint angles in the $i$th leg kinematic chain |

| | |
|---|---|
| ${}^{i,j}\mathbf{R}_{j+1}\left(\theta_{i,j}\right)$ | Rotation matrix of coordinate system $\left\{{}^{i}C_{j+1}\right\}$ with respect to coordinate system $\left\{{}^{i}C_{j}\right\}$ in the $i$th leg kinematic chain, with $i \in \{1,2,3,4,5,6\}$ and $j \in \{1,2,3\}$ |
| ${}^{i,j}\mathbf{T}_{j+1}\left(\theta_{i,j}\right)$ | Homogeneous transformation matrix of the coordinate system $\left\{{}^{i}C_{j+1}\right\}$ with respect to the coordinate system $\left\{{}^{i}C_{j}\right\}$ in the $i$th leg kinematic chain, with $i \in \{1,2,3,4,5,6\}$ and $j \in \{1,2,3\}$ |
| SO(m) | Special orthonormal group of real $(m \times m)$ matrices with orthonormal columns and determinant equal to 1 |

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
