# Peer review of "Adaptive Robust Autonomous Obstacle Traversal Controller for Novel Six-Track Robot"

_machines, doi:10.3390/machines11030378_

Round 1
Reviewer 1 Report
The paper is devoted to describing the new approach to constructing an adaptive controller for a six-track robot. The controller is derived using kinetical model approach, and takes into account also the possible slipping effect of the flippers. Considerations are supported by experiments, both in silico and in vivo, confirming its practical usability in some steps, trenches and sloped terrains scenarios.
The manuscript is clearly written, properly structured with potentially applicable results. I recommend its publication in MDPI Machines journal.
Reviewer 2 Report
1. The drive motor, control board, control algorithm block diagram and software and hardware structure shall be presented in the text;
2. Whether the spring on each leg affects the motion and control performance of the robot, which needs to be explained in the text;
3. The subfigures of Figure 10 and Figure 12 are lack of necessary titles;
4. Whether the minus sign of Formula 41 should be a positive sign;
5. Whether the kinematics analytical models established in this paper have also been verified in the simulation, and how accurate is it;
Reviewer 3 Report
The topic is promising, however, it is suggested the following aspects:
1. All figures are bad quality, except for figure 4. Therefore, the Authors must improve all figures in high resolution.
2. Add in to the abstract section, the value of the important metric 0.029 m of average trajectory error of the flipper.
3. In the conclusion section discuss the permissibility of error in every studied case, detailing limits and scope.
